# Oracle-Efficient Differentially Private Learning with Public Data

**Adam Block**
Department of Mathematics
MIT
Cambridge, MA 02139
ablock@mit.edu

**Mark Bun**
Department of Computer Science
Boston University
Boston, MA 02215
mbun@bu.edu

**Rathin Desai**
Department of Computer Science
Boston University
Boston, MA 02215
rathin@bu.edu

**Abhishek Shetty**
Department of Computer Science
University of California, Berkeley
Berkeley, CA 94720
shetty@berkeley.edu

**Zhiwei Steven Wu**
School of Computer Science
Carnegie Mellon University
Pittsburgh, PA 15213
zstevenwu@cmu.edu

## Abstract

Due to statistical lower bounds on the learnability of many function classes under privacy constraints, there has been recent interest in leveraging public data to improve the performance of private learning algorithms. In this model, algorithms must always guarantee differential privacy with respect to the private samples while also ensuring learning guarantees when the private data distribution is sufficiently close to that of the public data. Previous work has demonstrated that when sufficient public, unlabelled data is available, private learning can be made statistically tractable, but the resulting algorithms have all been computationally inefficient. In this work, we present the first computationally efficient, algorithms to provably leverage public data to learn privately whenever a function class is learnable non-privately, where our notion of computational efficiency is with respect to the number of calls to an optimization oracle for the function class. In addition to this general result, we provide specialized algorithms with improved sample complexities in the special cases when the function class is convex or when the task is binary classification.

## 1 Introduction

Differential privacy (DP) [Dwork et al., 2006] is a standard guarantee of individual-level privacy for statistical data analysis. Algorithmic research on differential privacy aims to understand what statistical tasks are compatible with the definition, and at what cost, e.g., in terms of sample complexity or computational efficiency. Unfortunately, it is known that some tasks may become more expensive or outright impossible to conduct with differential privacy. For example, in the setting of binary classification, there is no differentially private algorithm for solving the simple problem of learning a one-dimensional classifier over the real numbers [Bun et al., 2015, Alon et al., 2019].

38th Conference on Neural Information Processing Systems (NeurIPS 2024).

Motivated in part by such barriers to full-fledged private learning, many papers have considered relaxing the model to allow the use of auxiliary "public" data Balcan and Feldman [2013], Bassily et al. [2019, 2020b, 2022, 2023], Kairouz et al. [2021], Amid et al. [2022], Lowy et al. [2023]. Such data may be available if individuals can voluntarily opt-in to share or sell their information to enable a particular task. Alternatively, a data analyst might have background knowledge about the underlying data distribution from the results of previous analyses, or hold a plausible generative model for it. These situations are captured by semi-private learning, first discussed by Balcan and Feldman [2013], formally introduced by Beimel et al. [2014] and subsequently studied by Bassily et al. [2019], Hopkins et al. [2024]. In this model, a learning algorithm is given $n$ "private" samples from a joint distribution $\mathcal{D}$ over example-label pairs, as well as $m$ *unlabeled* "public" samples from the same marginal distribution over examples. The algorithm must be differentially private with respect to its private dataset, but can depend arbitrarily on its public samples. For learning a binary classifier over a class $\mathcal{F}$ with a VC-dimension $\mathsf{vc}(\mathcal{F})$, these papers showed that in the presence of $O(\mathsf{vc}(\mathcal{F}))$ public unlabeled samples, every concept class $\mathcal{F}$ is agnostically learnable with $O(\mathsf{vc}(\mathcal{F}))$ private labeled samples, matching what is achievable without privacy guarantees.

While these results essentially resolve the statistical complexity of semi-private learning, they do not address the question of computational efficiency. These algorithms proceed by drawing enough public samples to construct a cover for the class $\mathcal{F}$ with respect to the target marginal distribution on examples, and then using the exponential mechanism [McSherry and Talwar, 2007] to select a hypothesis from this cover that fits the private dataset. As the size of this cover is exponential in $\mathsf{vc}(\mathcal{F})$, constructing it explicitly is computationally expensive. This paper aims to address the following question: *Is such computational overhead really necessary if $\mathcal{F}$ exhibits additional structure that make non-private learning tractable?*

In this work, we give new semi-private learners that are efficient whenever fast non-private algorithms are available. More specifically, our main result is generic semi-private algorithms for regression and classification that are *oracle-efficient* in that they run in polynomial time given an oracle solving the *non-private* empirical risk minimization problem for $\mathcal{F}$, and have sample complexity polynomial in the usual parameters such as Gaussian complexity and VC dimension.

**Theorem 1** (Informal version of Theorem 2). *Fix a function class $\mathcal{F} : \mathcal{X} \to [-1, 1]$. Then there is an oracle-efficient, $(\varepsilon, \delta)$-differentially private algorithm (Algorithm 2) using $\mathsf{poly}(\sup_m \overline{\mathcal{G}}_m(\mathcal{F}))$ labeled private samples (where, $\overline{\mathcal{G}}_m(\mathcal{F})$ denotes the Gaussian complexity of function class $\mathcal{F}$), unlabeled public samples, and calls to an empirical risk minimization oracle for $\mathcal{F}$ that learns an approximately optimal predictor $\hat{f} \in \mathcal{F}$.*

While Theorem 1 captures extremely broad learning settings, the polynomials governing its sample complexity are rather large. We identify several important cases in which the sample complexity can be improved and the number of oracle calls is only 2. In the case where the function class $\mathcal{F}$ is convex, we give a variant of Algorithm 2, inspired by follow-the-regularized-leader, with significantly improved sample complexity as a function of the desired error (Theorem 3, Algorithm 3). Finally, in the special case of binary classification (i.e., Boolean $\mathcal{F}$ under the 0-1 loss), we give a completely different oracle-efficient algorithm with improved sample complexity (Theorem 4, Algorithm 4), which requires the private sample size to grow at the rate of $O((\mathsf{vc}(\mathcal{F}))^2)$. Prior work of Bassily et al. [2018] gave an oracle-efficient algorithm in this setting with somewhat better sample complexity than ours, based on a reduction to private classification. Meanwhile, our algorithm has the advantage of being able to guarantee pure (rather than only approximate) differential privacy algorithm, as well as making only two oracle calls as opposed to the polynomially many as a function of $\varepsilon$ and the target accuracy. Our results in the binary classification setting can also be viewed as an extension of Neel et al. [2019], which gives oracle-efficient private learners for structured function classes $\mathcal{F}$ that have a small *universal identification set* [Goldman et al., 1993]. Our results relax this stringent combinatorial condition by leveraging a small public unlabelled dataset, which allows us to design an oracle-efficient private learner for any function class $\mathcal{F}$ with bounded VC-dimension.

In fact, our results also address a somewhat more general setting than the semi-private model described so far. Specifically, our results automatically handle the setting where there may be a bounded distribution shift between the public and private data. In particular, all of our results hold as long as the public unlabelled data distribution and the private marginal distribution over the fea-

ture space have a density ratio bounded by $\sigma$.[1] The standard semi-private setting corresponds to the special case where $\sigma = 1$. Taking this view, we can interpret our results as oracle-efficient private learning in the *smoothed learning* setting. Our algorithms achieve accurate learning provided that the private marginal distribution does not deviate too much from a public reference distribution. However, our privacy guarantees hold even if the private data distribution has unbounded distribution shift from the public data.

## 1.1 Related Work

Our work brings together ideas and techniques from multiple literatures.

**Oracle efficiency in private and online learning.** Our notion of oracle-efficiency is standard in (theoretical) machine learning to model reductions in a world where worst-case hardness abounds, but optimization heuristics (e.g., integer programming solvers, non-convex optimization) often enjoy success. Within the differential privacy literature, oracle-efficient algorithms are known for binary classification with classes $\mathcal{F}$ that admits small universal identification sets [Neel et al., 2019], synthetic data generation [Gaboardi et al., 2014, Nikolov et al., 2013, Neel et al., 2019, Vietri et al., 2020], and certain types of non-convex optimization problems [Neel et al., 2020]. Oracle-efficiency is also well-established approach in *online learning* [Kalai and Vempala, 2005, Hazan and Koren, 2016, Kozachinskiy and Steifer, 2023, Haghtalab et al., 2022a, Block and Simchowitz, 2022, Block and Polyanskiy, 2023, Block et al., 2023a,b], an area with deep connections to differential privacy [Alon et al., 2019, Abernethy et al., 2019, Bun et al., 2020, Ghazi et al., 2021]. Indeed, our new semi-private learning algorithm Algorithm 2 adapts a follow-the-perturbed-leader inspired algorithm [Block et al., 2022] from the setting of *smoothed* online learning.

**DP learning and release with public (unlabelled) data** Our results contribute to a long line of theoretical work that leverages public data for private data analysis. In particular, our work provides general computationally efficient algorithms (in the oracle efficiency sense) for semi-private learning [Beimel et al., 2014]. In addition to the work in this direction we discussed above, several recent papers Bassily et al. [2022, 2023] developed efficient algorithms for private learning with domain adaptation from a public source. That work accommodates a more general notion of distribution shift than ours, but makes essential use of *labeled* public data, as well as handling only restricted concept classes or loss functions. There has also been work that leverages public data to remove statistical barriers in private query release [Bassily et al., 2020a] and density estimation Bie et al. [2022], Ben-David et al. [2023]. Papernot et al. [2018], Yu et al. [2022], Golatkar et al. [2022], Zhou et al. [2021] and Liu et al. [2021a,b] give empirical guarantees to the problem of private learning and private synthetic data from public samples respectively.

**Smoothed Analysis in Online Learning.** Smoothed analysis was pioneered in Spielman and Teng [2004] for the purpose of explaining the empirical success of algorithms whose worst-case behavior is provably intractable. More recently, the framework has come to online learning [Rakhlin et al., 2011, Haghtalab et al., 2020, 2022b, Block et al., 2022, Haghtalab et al., 2022a, Block and Simchowitz, 2022, Block et al., 2023a,b] in order to circumvent the strong statistical [Rakhlin et al., 2015] and computational [Hazan and Koren, 2016] lower bounds that worst-case data can induce. The assumption of smoothness has also been used in learning more broadly [Durvasula et al., 2023, Cesa-Bianchi et al., 2023] and its assumptions have been relaxed [Block and Polyanskiy, 2023].

## 2 Preliminaries

In this section, we formally introduce our setting. Let $\mathcal{X}$ denote the feature space and $\mathcal{Y}$ be the label space. In general, we consider $\mathcal{Y} = [-1, 1]$, but in the special case of binary classification setting, we have $\mathcal{Y} = \{0, 1\}$. In general, we study learning algorithms $\mathcal{A}$ that map a dataset $\mathcal{D}$ with $n$ examples from $\mathcal{X} \times \mathcal{Y}$ to a predictor in a function class $\mathcal{F}$. We require $\mathcal{A}$ to satisfy *differential privacy*, defined below.

---

[1]In fact, this condition can be further relaxed to only assuming the two distributions have a bounded $f$-divergence.

**Definition 1** (Differential Privacy Dwork et al. [2006]). Let $\mathcal{A} : (\mathcal{X} \times \mathcal{Y})^n \to \mathcal{F}$ be a randomized algorithm and $\mathcal{D}, \mathcal{D}' \in (\mathcal{X} \times \mathcal{Y})^n$ be data sets. We say that $\mathcal{D}$ and $\mathcal{D}'$ are *neighboring* if $|\mathcal{D} \setminus \mathcal{D}'| = |\mathcal{D}' \setminus \mathcal{D}| \leq 1$, i.e. they differ in at most one datum. We say that $\mathcal{A}$ is $(\varepsilon, \delta)$-*differentially private* if for all neighboring datasets $\mathcal{D}, \mathcal{D}'$, and for all measurable $\mathcal{G} \subset \mathcal{F}$, it holds that $\mathbb{P}(\mathcal{A}(\mathcal{D}) \in \mathcal{G}) \leq e^{\varepsilon} \cdot \mathbb{P}(\mathcal{A}(\mathcal{D}') \in \mathcal{G}) + \delta$. If $\delta = 0$, we say that $\mathcal{A}$ is $\varepsilon$-(purely) differentially private.

As defined, it is trivial to construct algorithms that are differentially private by outputting functions independent of the data set; for an algorithm to be useful, however, we also require that it learns in a meaningful sense. Thus, in the context of learning, we consider the following accuracy desideratum.

**Definition 2.** Let $\mathcal{A} : (\mathcal{X} \times \mathcal{Y})^n \to \mathcal{F}$ be a randomized algorithm. We say that $\mathcal{A}$ is an $(\alpha, \beta)$-learner with respect to a measure $\nu$ on $\mathcal{X} \times \mathcal{Y}$ and loss function $L : \mathcal{F} \to [-1, 1]$ if, for $\mathcal{D}$ sampled independently from $\nu$, it holds that $\mathbb{P}(L(\mathcal{A}(\mathcal{D})) \leq \inf_{f \in \mathcal{F}} L(f) + \alpha) \geq 1 - \beta$. For regression problems, we consider the loss function $L$ to be induced by a function $\ell : \mathcal{Y} \times \mathcal{Y} \to [0, 1]$, convex and $\lambda$-Lipschitz in the first argument, such that $L(f) = \mathbb{E}_{(X,Y) \sim \nu} [\ell(f(X), Y)]$.

For simplicity, we will denote the empirical loss on a data set $\mathcal{D}$ as $L_{\mathcal{D}}(f) = \frac{1}{n} \cdot \sum_{(X_i, Y_i) \in \mathcal{D}} \ell(f(X_i), Y_i)$. We emphasize that in contradistinction to the standard notion of PAC-learnability [Valiant, 1984], our requirement is weaker in that we only require *distribution-dependent* learning, i.e., the algorithm $\mathcal{A}$ is allowed to depend on $\nu$ in some to-be-specified way. This is necessary in our setting as it is well known that distribution-independent differentially private PAC learning is possible only for very restricted classes of functions $\mathcal{F}$ with bounded Littlestone dimension [Alon et al., 2019, Bun et al., 2020]. To make private learning statistically tractable for broader classes of functions, we consider the following restriction on $\nu$:

**Definition 3.** Given a measure $\mu \in \Delta(\mathcal{X})$ and a parameter $\sigma \in (0, 1]$, we say that $\nu_x$ is $\sigma$-smooth with respect to $\mu$ if $\left\| \frac{d\nu_x}{d\mu} \right\|_{\infty} \leq \frac{1}{\sigma}$. We suppose that the learner has access to $m$ samples $Z_1, \ldots, Z_m \sim \mu$ that are independent of each other and the training data $\mathcal{D}$ and thus $\mathcal{A}$ may depend on these samples.

We remark that Definition 3 can be significantly relaxed by assuming only that $D_f(\nu_x || \mu) \leq \frac{1}{\sigma}$ as in Block and Polyanskiy [2023], where $D_f(\cdot || \cdot)$ is a sufficiently strong $f$-divergence[2]. In this case, the statistical rates presented below will be worse and depend on $f$, but the algorithms and privacy guarantees will remain unchanged. Critically, we do not require that our algorithms are private with respect to $Z_1, \ldots, Z_m$, which we treat as public, unlabelled data. The key reason that this public data helps us circumvent the lower bounds is that it gives us access (albeit indirectly) to a small subclass of the hypothesis set that still has approximately good hypotheses. Since our primary focus is to design computationally efficient private learners, we cannot directly handle either the original hypothesis class or the small proxy that the public data gives us access. Instead we suppose access to the following ERM oracle:

**Definition 4.** Given a function class $\mathcal{F} : \mathcal{X} \to \mathbb{R}$, a data set $\mathcal{D} = \{x_1, \ldots, x_m\} \subset \mathcal{X}$ and loss functions $\ell_1, \ldots, \ell_m : \mathbb{R} \to \mathbb{R}$, we define the empirical risk minimization oracle $\mathsf{ERM} : \mathcal{F} \to \mathbb{R}$ such that $\mathsf{ERM}(\mathcal{F}, \mathcal{L}_{\mathcal{D}}) \in \arg\min_{f \in \mathcal{F}} \mathcal{L}_{\mathcal{D}}(f)$, where $\mathcal{L}_{\mathcal{D}}(f) = \sum_{x_i \in \mathcal{D}} \ell_i(f(x_i))$.

ERM oracles are standard computational models in many learning domains such as online learning [Kalai and Vempala, 2005, Hazan and Koren, 2016, Block et al., 2022, Haghtalab et al., 2022a] and Reinforcement Learning [Foster and Rakhlin, 2020, Foster et al., 2021, Mhammedi et al., 2023b,a]. Assuming access to ERM allows us to disentangle the computational challenges of optimizing over specific function classes from the specific challenge of differentially private learning as well as to avoid the well-known intractability results for nonconvex optimization [Blum and Rivest, 1988] that do not accurately reflect the realities of modern optimization techniques (e.g., integer program solvers, SGD). We note that our algorithms also work in the case of ERM oracles with additive error by minor modification to the analysis similar to the one in [Block et al., 2022]. We remark that applying Neel et al. [2019, Theorem 8] gives a black-box robustification procedure for purely private, oracle-efficient algorithms, which ensures that the privacy guarantees continue to hold even when the oracle may fail to optimize the objective. In particular, Algorithms 3 and 4 below, when run in their pure DP forms can be made robust at a minimal cost on accuracy. We defer to Neel et al. [2019] for further discussion on this topic.

---

[2]These divergences include the well-known KL divergence and Renyi divergence. For a comprehensive introduction to $f$-divergences, see Polyanskiy and Wu [2022+].

---
**Algorithm 1:** Perturb: An algorithm for perturbing a function with noise on public data.
---
1: **Input** Function $\bar{f} \in \mathcal{F}$, distribution $\mathcal{Q} \in \Delta(\mathbb{R})$, scale $\gamma \geq 0$, public data $\widetilde{\mathcal{D}}_x = \{Z_1, \ldots, Z_m\}$.
2: **Sample** $\zeta_1, \ldots, \zeta_m \sim \mathcal{Q}$.
3: **Define** $R(f) = \left\| f - \bar{f} - \gamma \cdot \zeta \right\|_m^2$.
4: **Output** $\widehat{f} = \mathsf{ERM}(R, \mathcal{F})$.
---

It is well-known that even absent differential privacy guarantees, learning arbitrary function classes is impossible; we now introduce the notions of complexity that are relevant to our results. We begin with the standard notion of VC dimension:

**Definition 5.** Let $\mathcal{F} : \mathcal{X} \to \{0, 1\}$ be a function class. We say that a set of points $x_1, \ldots, x_d \in \mathcal{X}$ shatters $\mathcal{F}$ if for all $\varepsilon_{1:d} \in \{0, 1\}^d$, there is some $f_\varepsilon$ such that $f_\varepsilon(x_i) = \varepsilon_i$ for all $i$. The VC dimension of $\mathcal{F}$, denoted $\mathsf{vc}(\mathcal{F})$, is the largest $d$ such that there exists a set of $d$ points shattering $\mathcal{F}$.

In addition to VC dimension, we also use the Gaussian complexity of a function class:

**Definition 6.** Let $\mathcal{F} : \mathcal{X} \to [-1, 1]$ be a function class and $x_1, \ldots, x_m \in \mathcal{X}$ be arbitrary points. We let $\omega_m : \mathcal{F} \to \mathbb{R}$ be the canonical Gaussian process on $\mathcal{F}$, i.e.,

$$\omega_m(f) = \frac{1}{\sqrt{m}} \cdot \sum_{i=1}^m \xi_i \cdot f(x_i), \tag{1}$$

where $\xi_i$ are independent standard Gaussians. We define the (data-dependent) Gaussian complexity of $\mathcal{F}$ to be $\mathbb{E}\left[\sup_{f \in \mathcal{F}} \omega_m(f)\right]$, the average Gaussian complexity as $\mathcal{G}_m(\mathcal{F}) = \mathbb{E}_Z \mathbb{E}[\sup_{f \in \mathcal{F}} \omega_m(f)]$, and the worst-case Gaussian complexity of $\mathcal{F}$ to be $\overline{\mathcal{G}}_m(\mathcal{F}) = \sup_{x_1, \ldots, x_m \in \mathcal{X}} \mathbb{E}\left[\sup_{f \in \mathcal{F}} \omega_m(f)\right]$.

Both $\mathsf{vc}(\mathcal{F})$ and $\mathcal{G}_m(\mathcal{F})$ are well known measures of complexity from learning theory and their relationships to other notions of complexity like covering number are well-understood [Mendelson and Vershynin, 2003, Wainwright, 2019, Van Handel, 2014]. In particular, it is well-known that $\mathcal{G}_m(\mathcal{F}) = O(\sqrt{\mathsf{vc}(\mathcal{F})})$ [Dudley, 1969, Mendelson and Vershynin, 2003] and that standard PAC-learning is possible if and only if $\overline{\mathcal{G}}_m(\mathcal{F}) = o(\sqrt{m})$ [Wainwright, 2019, Van Handel, 2014]. We remark that different texts use different scalings for $\mathcal{G}_m(\mathcal{F})$, with some replacing the $m^{-1/2}$ factor in (1) with $m^{-1}$ and others omitting it entirely; our choice of scaling is motivated by the fact that a natural complexity measure for many (Donsker [Wainwright, 2019]) function classes that our algorithms depend on is $\sup_m \overline{\mathcal{G}}_m(\mathcal{F})$, which is most compactly represented with the present scaling.

**Notation.** We always reserve $\mathbb{P}$ and $\mathbb{E}$ for probability and expectation with respect to measures that are clear from the context. We denote by $\Delta(\mathcal{X})$ the space of measures on some $\mathcal{X}$ and for any $\mu \in \Delta(\mathcal{X})$ we let $\|\cdot\|_\mu$ denote the $L^2(\mu)$ norm, i.e., $\|f\|_\mu^2 = \mathbb{E}_{Z \sim \mu}[f(Z)^2]$. Similarly, for $m$ points $Z_1, \ldots, Z_m \in \mathcal{X}$, we let $\|\cdot\|_m$ denote the empirical $L^2$ norm on these points so that $\|f\|_m^2 = m^{-1} \cdot \sum_{i=1}^m f(Z_i)^2$. We reserve $\omega_m$ for the canonical empirical Gaussian process on $\mathcal{F}$ as in (1) and $\mathcal{L}$ for a functional on $\mathcal{F}$.

## 3 Algorithms for Differentially Private Learning

In this section, we provide a general template for constructing differentially private learning algorithms with public data and instantiate this template with two oracle-efficient algorithms. Our first algorithm applies to arbitrary bounded function classes, whereas the second algorithm only applies to convex classes but has an improved sample complexity. Our general template is broken into the following two steps: (i) Use ERM (cf. Definition 4) and the public data to construct an initial estimate $\bar{f}$ that is a good learner and satisfies stability with respect to $\|\cdot\|_m$; (ii) Output $\widehat{f}$ as the function that minimizes $\left\| f - \bar{f} - \gamma \cdot \zeta \right\|_m$, where $\gamma \geq 0$ is a scale and $\zeta = (\zeta_1, \ldots, \zeta_m)$ is a vector of independent random variables sampled according to some distribution $\mathcal{Q}$. The second step is accomplished through Algorithm 1 and is the same across our algorithms. The first step, however,

---

**Algorithm 2:** Oracle Efficient Private Learner (Perturbation)

---

1: **Input** Oracle ERM, perturbation parameter $\eta > 0$, public data set $\widetilde{\mathcal{D}}_x = \{Z_1, \ldots, Z_m\}$, private data set $\mathcal{D} = \{(X_i, Y_i) | 1 \leq i \leq n\}$, function class $\mathcal{F}$, loss function $\ell$, noise level $\gamma > 0$, number of iterations $J \in \mathbb{N}$, noise distribution $\mathcal{Q} \in \Delta(\mathbb{R})$.

2: **for** $j = 1, 2, \ldots, J$ **do**

3:     **Sample** $\xi_1^{(j)}, \ldots, \xi_m^{(j)} \sim \mathcal{N}(0, 1)$.

4:     **Define** $\omega_m^{(j)} : \mathcal{F} \to \mathbb{R}$ such that

$$\omega_m^{(j)}(f) = \frac{1}{\sqrt{m}} \cdot \sum_{i=1}^m \xi_i \cdot f(Z_i). \tag{2}$$

5:     **Define** $\mathcal{L}^{(j)} : \mathcal{F} \to \mathbb{R}$ such that $\mathcal{L}^{(j)}(f) = \sum_{(X_i, Y_i) \in \mathcal{D}} \ell(f(X_i), Y_i) + \eta \cdot \omega_m^{(j)}(f)$.

6:     **Define** $\bar{f}_j = \mathsf{ERM}(\mathcal{L}^{(j)}, \mathcal{F})$
    **end**

7: **Define** $\bar{f} = \frac{1}{J} \cdot \sum_{j=1}^J \bar{f}_j$.

8: **Output** $\widehat{f} = \mathsf{Perturb}(\bar{f}, \mathcal{Q}, \gamma, \widetilde{\mathcal{D}}_x)$             ▷ By running Algorithm 1

---

is algorithm-specific and is the primary factor affecting the sample complexity. The intuition for our template is as follows. We need to show that $\widehat{f}$ is both a learner and is differentially private. To see why the template produces a good learner, note that if $Z_i \sim \mu$ are independent and $m$ is sufficiently large, then $\|\cdot\|_m \approx \|\cdot\|_\mu$. Thus if $\gamma$ is small, then $\left\| \bar{f} - \widehat{f} \right\|_\mu \ll 1$ and $\left| \mathbb{E}_\mu \left[ L(\widehat{f}) \right] - \mathbb{E}_\mu[L(\bar{f})] \right| \ll 1$ whenever $\ell$ is Lipschitz. By smoothness, a similar guarantee holds for expectations with respect to $\nu$ and thus $\widehat{f}$ is a good learner. To see why $\widehat{f}$ is differentially private, note that by choosing $\mathcal{Q}$ to be a standard Gaussian, we can ensure that the likelihood ratios of choosing $\widehat{f}$ given $\bar{f}$ versus $\bar{f}'$ are controlled by $\left\| \bar{f} - \bar{f}' \right\|_m$. Thus, if $\bar{f}$ is stable with respect to $\|\cdot\|_m$, then $\widehat{f}$ will be private. The intuition of the stability of $\bar{f}$ is discussed in Section 4.

We now make the above intuition precise by instantiating this template in our most general setting in Algorithm 2. We construct $\bar{f}$ by running ERM on a perturbed version of the empirical risk minimization problem and then averaging. Specifically, for $j \in [J]$, we define $\mathcal{L}^{(j)} : \mathcal{F} \to \mathbb{R}$ as a sample path of a noncentred Gaussian process in (2) and let $\bar{f}_j$ denote the minimizer of $\mathcal{L}^{(j)}$ over $\mathcal{F}$. We then output $\bar{f}$ as the average of $\bar{f}_1, \ldots, \bar{f}_J$. We present motivation for the particular choice of $\bar{f}$, as well as the analogue in Algorithm 3, in the subsequent section. The following theorem shows that if $\mathcal{Q}$ is chosen correctly, this algorithm is an oracle-efficient differentially private learner whenever $\nu_x$ is $\sigma$-smooth with respect to $\mu$.

**Theorem 2.** *Suppose that $\mathcal{F} : \mathcal{X} \to [-1, 1]$ is a function class and $\mu \in \Delta(\mathcal{X})$ is a measure such that $\inf_{f \in \mathcal{F}} \|f\|_\mu \geq \frac{2}{3}$. Let $\ell : [-1, 1] \times [-1, 1] \to [0, 1]$ be a loss function that is convex and $\lambda$-Lipschitz in its first argument. If $\mathcal{Q} = \mathcal{N}(0, 1)$ in Algorithm 1, then for any $\varepsilon, \delta, \alpha, \beta \in (0, 1)$, there are choices of $\eta, \gamma > 0$ and $J, m \in \mathbb{N}$, all polynomial in problem parameters and given in Appendix C.6, such that if*

$$n = \mathsf{poly}\left( \sup_m \overline{\mathcal{G}}_m(\mathcal{F}), \log\left(\frac{1}{\delta}\right), \log\left(\frac{1}{\beta}\right), \lambda \right) \cdot \varepsilon^{-3} \cdot \alpha^{-14},$$

*then the $\widehat{f}$ returned by Algorithm 2 is $(\varepsilon, \delta)$-differentially private. If $\nu_x$ is $\sigma$-smooth with respect to $\mu$, then $\widehat{f}$ is an $(\alpha, \beta)$-learner with respect to $\nu_x$ and $\ell$.*

We emphasize that Algorithm 2 is *always differentially private*, independent of $\nu$; however, our algoirthm is only a good learner if $\nu_x$ is smooth with respect to $\mu$. We remark that all of the conditions in Theorem 2 are standard with the exception of the assumption that $\|f\|_\mu \geq 2/3$ for all $f \in \mathcal{F}$. This condition is easy to ensure by setting $\widetilde{\mu} = \frac{\mu + 2 \cdot \delta_{z^\star}}{3}$, where $z^\star$ is a distinguished point such that $f(z^\star) = 1$ for all $f \in \mathcal{F}$; note that this process deflates $\sigma$ at most by a factor of 3 while ensuring the lower bound on the norm of $f$. Replacing $\mu$ by $\widetilde{\mu}$ then suffices to ensure that Theorem 2 holds.

**Algorithm 3:** Oracle Efficient Private Learner (Regularization)

---

1: **Input** Oracle ERM, perturbation parameter $\eta > 0$, public data set $\widetilde{\mathcal{D}}_x = \{Z_1, \ldots, Z_m\}$, private data set $\mathcal{D} = \{(X_i, Y_i) | 1 \leq i \leq n\}$, function class $\mathcal{F}$, loss function $\ell$, noise level $\gamma > 0$, number of iterations $J \in \mathbb{N}$, noise distribution $\mathcal{Q} \in \Delta(\mathbb{R})$.
2: **Define** $\mathcal{L} : \mathcal{F} \to \mathbb{R}$ such that

$$\mathcal{L}(f) = \sum_{(X_i, Y_i) \in \mathcal{D}} \ell(f(X_i), Y_i) + \eta \cdot \|f\|_m^2. \tag{3}$$

3: **Define** $\bar{f} = \mathsf{ERM}(\mathcal{L}, \mathcal{F})$
4: **Output** $\widehat{f} = \mathsf{Perturb}(\bar{f}, \mathcal{Q}, \gamma, \widetilde{\mathcal{D}}_x)$          ▷ By running Algorithm 1

---

We further remark that it is classical that the complexity notion $\sup_m \overline{\mathcal{G}}_m(\mathcal{F})$ is upper bounded by $\sqrt{\mathsf{vc}(\mathcal{F})}$ for binary function classes and $\sqrt{\log(|\mathcal{F}|)}$ for finite classes [Wainwright, 2019], ensuring that the proven sample complexity is polynomial in all standard notions of function class complexity. For even more complex function classes, where $\overline{\mathcal{G}}_m(\mathcal{F}) = \omega(1)$, similar results hold, although with worse rates; further dicussion, as well as the precise polynomial dependence of hyperparameters and sample complexity, can be found in Appendix C.

While Algorithm 2 succeeds in our desiderata under general assumptions, the sample complexity is a large polynomial of the desired accuracy. Indeed, the sample complexity of Algorithm 2 scales like $O\left(\mathsf{vc}(\mathcal{F}) \cdot \varepsilon^{-3} \cdot \alpha^{-14}\right)$, which is significantly worse than the $O\left(\mathsf{vc}(\mathcal{F}) \cdot \alpha^{-2}\right)$ sample complexity that a non-private algorithm such as ERM can achieve [Wainwright, 2019] or even the $O\left(\mathsf{vc}(\mathcal{F}) \cdot \alpha^{-2} \cdot \varepsilon^{-2}\right)$ sample complexity achievable by private, inefficient algorithms with public data [Bassily et al., 2020b]. Furthermore, we are unable to achieve a pure differential privacy guarantee with this algorithm. We now address both issues by providing an improved algorithm in the special case that the function class $\mathcal{F}$ is *convex*. While we still use Algorithm 1 as a subroutine, in Algorithm 3, motivated by the difference between Follow the Perturbed Leader (FTPL) and Follow the Regularized Leader (FTRL) [Kalai and Vempala, 2005, Cesa-Bianchi and Lugosi, 2006] in online learning, we modify the way in which we choose our initial estimator $\bar{f}$. In particular, we eliminate the averaging step and redefine $\omega$ to be a strongly convex *regularizer* instead of a Gaussian Process *perturbation*. More specifically, we define $\mathcal{L}$ in (3) as the empirical loss regularized by $\|\cdot\|_m^2$ and output $\bar{f} = \mathsf{ERM}(\mathcal{L}, \mathcal{F})$. We have the following result:

**Theorem 3.** *Suppose that $\mathcal{F} : \mathcal{X} \to [-1, 1]$ is a convex function class and $\ell : [-1, 1] \times [-1, 1] \to [0, 1]$ is convex and $\lambda$-Lipschitz in its first argument. Suppose that $Z_1, \ldots, Z_m \sim \mu$ are independent and $\mathcal{Q} = \mathcal{N}(0, 1)$. Then there are $\eta, \gamma, m$ polynomial in problem parameters and given in Appendix C.6 such that, if*

$$n = \mathsf{poly}\left(\log\left(\beta^{-1}\right), \log\left(\delta^{-1}\right), \lambda\right) \cdot (\sup_m \overline{\mathcal{G}}_m(\mathcal{F}))^2 \cdot \varepsilon^{-1} \cdot \alpha^{-5}$$

*then the $\widehat{f}$ returned by Algorithm 3 is $(\varepsilon, \delta)$-differentially private. If $\nu_x$ is $\sigma$-smooth with respect to $\mu$, then $\widehat{f}$ is an $(\alpha, \beta)$-learner with respect to $\nu_x$ and $\ell$. Furthermore, if $\mathcal{Q} = Lap(1)$, and*

$$n = \mathsf{poly}\left(\sup_m \overline{\mathcal{G}}_m(\mathcal{F}), \log\left(\beta^{-1}\right), \lambda\right) \cdot \varepsilon^{-1} \cdot \alpha^{-6},$$

*then Algorithm 3 is $\varepsilon$-purely differentially private and an $(\alpha, \beta)$-PAC learner for any $\sigma$-smooth $\nu_x$.*

As in the case of Theorem 2, we can easily generalize Theorem 3 to apply to function classes $\mathcal{F}$ where $\overline{\mathcal{G}}_m(\mathcal{F}) = \omega(1)$ at the cost of worse polynomial dependence in the sample complexity. We again omit this case for the sake of simplicity. While the sample complexity of Algorithm 3 is a marked improvement over that of Algorithm 2, it remains a far cry from the desired $O\left(\alpha^{-2}\right)$ rates of non-private learning that computationally *inefficient* private algorithms leveraging public data are able to achieve [Bassily et al., 2019]; we leave the interesting question of producing an oracle-efficient private algorithm with optimal sample complexity to future work.

Finally, we remark that even in the case where $\mathcal{F}$ is not convex, Algorithm 3 can be applied to $\mathrm{conv}(\mathcal{F})$, the convex hull of $\mathcal{F}$, if we assume the learner has access to $\mathsf{ERM}'$, a stronger ERM oracle

that can optimize over $\mathrm{conv}(\mathcal{F})$. In this case, Theorem 3 supercedes Theorem 2 as it is easy to see that $\overline{\mathcal{G}}_m(\mathcal{F}) = \overline{\mathcal{G}}_m(\mathrm{conv}(\mathcal{F}))$ and thus the sample complexity of Algorithm 3 is strictly better than that of Algorithm 2 and the pure differential privacy result applies.

# 4    Analysis Techniques

In this section, we outline the proofs of our main results, with full details deferred to Appendix C. As is suggested by our template, the proof of the privacy part of Theorem 2 rests on two results: the first shows that if $\mathcal{Q}$ is a standard Gaussian (resp. exponential) then stability of $\bar{f}$ with respect to $\|\cdot\|_m$ can be translated into differential privacy. The second shows that $\bar{f}$ will be stable with respect to $\|\cdot\|_m$. Similarly, the proof that $\widehat{f}$ is a good learner first shows that $\bar{f}$ is a good learner and then that $\widehat{f}$ and $\bar{f}$ are close. We begin with the more technically novel parts and show that, under standard assumptions, Algorithms 2 and 3 result in $\bar{f}$ that are stable in $\|\cdot\|_m$. In our proof of Theorem 2, we provide an improved analysis of the Gaussian anti-concentration result from Block et al. [2022], which may be of independent interest. We prove the stability of Algorithm 3 using a technique common in online learning. We then show that stability in $\|\cdot\|_m$ can be boosted to a differential privacy guarantee using the Gaussian and Laplace Mechanisms [Dwork et al., 2006]. Finally, we apply standard learning theoretic techniques to show that $\widehat{f}$ is a good learner.

## 4.1    Stability Analysis

In this section, we explain how to prove that Algorithms 2 and 3 are stable with respect to $\|\cdot\|_m$. Our stability results further cement the connections between differential privacy and online learning noted in Abernethy et al. [2019] as both algorithms are primarily motivated by online learning techniques. We begin by describing the stability analysis of Algorithm 2. The key lemma underlying the stability of Algorithm 2 is an improved version of a Gaussian anti-concentration result from Block et al. [2022], which may be of independent interest.

**Proposition 1.** *Let $\mathcal{F}$ denote a subspace of the unit ball with respect to a norm $\|\cdot\|$ induced by an inner product $\langle \cdot, \cdot \rangle$ and let $m, m' : \mathcal{F} \to \mathbb{R}$ be measurable functions such that $\|m - m'\|_\infty \leq \tau$. If $\omega$ is a centred Gaussian process on $\mathcal{F}$ with covariance kernel given by $\langle \cdot, \cdot \rangle$, $\Omega(f) = m(f) + \eta \cdot \omega(f)$, $\bar{f} = \mathrm{argmin}_{f \in \mathcal{F}} \Omega(f)$, and $\Omega'$ and $\bar{f}'$ are defined similarly, then for any $\rho, \tau > 0$, it holds that $\mathbb{P}\left( \|\bar{f} - \bar{f}'\| > \rho \right) \leq \frac{8\tau}{\rho^4 \kappa^2 \eta} \cdot \mathbb{E}\left[ \sup_{f \in \mathcal{F}} \omega(f) \right]$, where $\kappa^2 = \inf_{f \in \mathcal{F}} \mathbb{E}\left[ \omega(f)^2 \right]$.*

The proof proceeds in a similar way to that of Lemma 33 from Block et al. [2022], but involves a tighter analysis in several steps in order to improve the bound. The intuition for the result is straightforward: if $\bar{f}$ is the minimizer of the Gaussian process $\Omega$, then with reasonable probability, *almost minimizers* of $\Omega$ (as measured by the tolerance $\tau$) are within a radius $\rho$ of $\bar{f}$ as long as the Gaussian process is nontrivial in the sense that all indices $f$ have sufficiently high variance. Moreover, the quantitative control on the probability of this event depends in a natural way both on $\tau$ and $\rho$ as well as on the Gaussian process $\omega$: more complex spaces $\mathcal{F}$ and lower variance processes lead to a worse anti-concentration guarantee. Finally, we note that Proposition 1 is an improvement of Lemma 33 from Block et al. [2022] in that the quantitative bound on the probability of anti-concentration is tighter by polynomial factors in $\rho, \eta$, and $\kappa$.

Like essentially all anti-concentration results [Chernozhukov et al., 2015], Proposition 1 holds only with moderate probability in the sense that the guarantee is polynomial in the scale $\rho$; this fact is in contradistinction to *concentration* inequalities which tend to hold with high probability exponential in the scale. This discrepancy is precisely what motivates the averaging in Line 7 of Algorithm 2. Indeed, we can use Proposition 1 to show that if $\bar{f}_j$ is as in Line 6 of Algorithm 2 and $\bar{f}'_j$ is defined analogously with respect to $\mathcal{D}'$, then with moderate probability $\left\| \bar{f}_j - \bar{f}'_j \right\|_m$ is small. Using Jensen's inequality and a standard chernoff bound, we can then boost this moderate probability guarantee into a high probability guarantee to show that if $J$ is sufficiently large, then $\left\| \bar{f} - \bar{f}' \right\|_m$ is small with high probability. We formalize this argument in the following lemma:

**Lemma 1** (Stability of Algorithm 2). *Suppose that $\mathcal{F} : \mathcal{X} \to [-1, 1]$ is a function class and $\ell : [-1, 1]^{\times 2} \to [0, 1]$ is a bounded loss function. Suppose that $\mathcal{D}, \mathcal{D}'$ are neighboring datasets and let $\bar{f}$ be as in Line 5 of Algorithm 2 and $\bar{f}'$ be defined analogously with respect to $\mathcal{D}'$. Then*

*for any $\rho, \delta > 0$, with probability at least $1 - \delta$, over the Gaussian processes $\omega^{(j)}$, $\left\| \bar{f} - \bar{f}' \right\|_m \leq$*
$\frac{2}{(\eta \cdot n)^{1/3} \kappa^{2/3}} \cdot \left( \mathbb{E} \left[ \sup_{f \in \mathcal{F}} \omega_m(f) \right] \right)^{1/3} + \sqrt{\frac{\log\left(\frac{1}{\delta}\right)}{J}}.$

We note that the worse dependence on $\eta$ in Lemma 1 as compared to Proposition 1 arises from integrating the tail bound to obtain the control on $\left\| \bar{f}_j - \bar{f}'_j \right\|_m$ in expectation necessary to apply Jensen's inequality; details can be found in Appendix C.1.

We now turn to the stability of Algorithm 3. The proof is based on a technique borrowed from online learning and the analysis of the *Follow the Regularized Leader* (FTRL) algorithm [Gordon, 1999, Cesa-Bianchi and Lugosi, 2006].

**Lemma 2** (Stability of Algorithm 3)**.** *Suppose that $\ell$ is convex and $\lambda$-Lipschitz in its first argument. Let $\mathcal{D}, \mathcal{D}'$ denote neighboring data sets and let $\bar{f}$ denote the output of Line 3 in Algorithm 3 and $\bar{f}'$ be the analogous output evaluated on $\mathcal{D}'$. If $\mathcal{F}$ is convex, then, $\left\| \bar{f} - \bar{f}' \right\|_m \leq \frac{2}{\sqrt{\eta \cdot n}}$.*

The proof of Lemma 2 can be found in Appendix C.2 and rests on elementary properties of strongly convex functions. We note that relative to Lemma 1, the dependence on $\eta$ in Lemma 2 is improved, which in turn leads to the better sample complexity exhibited in Theorem 3. With stability of Algorithms 2 and 3 thus established, we proceed to analyze the effect of the output perturbation.

## 4.2 Output Perturbation Analysis

We now turn to the analysis of Algorithm 1. In order to boost a stability-in-norm guarantee into one for differential privacy while remaining a good learner, we require the output perturbation to be sufficiently small as to not not affect the learning guarantee of $\bar{f}$ while at the same time being sufficiently large as to ensure privacy. We balance these two competing objectives by tuning the variance of the added noise. This part of the analysis is relatively standard in the differential privacy literature [Chaudhuri et al., 2011, Neel et al., 2019], with the bound on the size of the output perturbation following from standard tail bounds on Gaussian and Laplace random vectors. The privacy guarantees are similarly standard and summarized in the following lemma:

**Lemma 3.** *Suppose $\bar{f} \in \mathcal{F}$ is the output of an algorithm $\mathcal{A} : \mathcal{D} \to \mathcal{F}$ that is $\rho$-stable with respect to $\|\cdot\|_m$, i.e., for any neighboring data set $\mathcal{D}'$, it holds that $\|\mathcal{A}(\mathcal{D}) - \mathcal{A}(\mathcal{D}')\|_m \leq \rho$. Then applying Algorithm 1 with $\mathcal{Q} = \mathcal{N}(0,1)$ to $\bar{f}$ results in an $(\varepsilon, \delta)$-private algorithm if $\frac{m}{2\gamma^2} \left( 1 + \gamma \cdot \sqrt{\log\left(\frac{1}{\delta}\right)} \right) \rho \leq \varepsilon$. Similarly, if $\mathcal{Q} = Lap(\gamma)$, then the algorithm is $\varepsilon$-purely private if $m^{3/2}/\gamma \cdot \rho \leq \varepsilon$.*

This standard result is proved in Appendix C.3. Note that, perhaps counterintuitively, the privacy loss increases with the public data. This relationship occurs because the algorithm is implicitly discretizing the function class, where more public data leads to a finer discretization; though finer discretizations lead to higher accuracy, they also leads to more privacy loss. Furthermore, note that even were the whole marginal distribution known, the privacy-accuracy tradeoff is dictated by the number of *labelled* samples, not $m$.

The balance between privacy and learning is quantified in the choices of $m$ and $\gamma$. If $\gamma$ is too large, then $\bar{f}$ will be private but a poor learner, whereas the opposite occurs if $\gamma$ is too small. Similarly, if $m$ is too large then privacy is reduced whereas if $m$ is too small then $\|\cdot\|_m$ is a poor approximation for $\|\cdot\|_\mu$.

## 4.3 Learning Guarantees and Concluding the Proof

By combining Lemma 1 (resp. Lemma 2) with Lemma 9, we can establish the privacy of Algorithm 2 (resp. Algorithm 3) as long as the tuning parameters $m, \gamma, \eta$, and $J$ are chosen correctly. We now sketch the proof that these algorithms comprise good learners in the sense of Definition 2. We break our proof into three components, the first two of which are standard learning theoretic results. The first lemma says that if $m \gg 1$, then $\|\cdot\|_m$ is a good approximation for $\|\cdot\|_\mu$:

**Lemma 4.** *Let $\mathcal{F} : \mathcal{X} \to [-1, 1]$ be a bounded function class and let $Z_1, \ldots, Z_m \sim \mu$ be independent samples. Then for any $\beta > 0$ it holds with probability at least $1 - \beta$ that for all $f \in \mathcal{F}$,*
$\|f\|_\mu \leq 2 \cdot \|f\|_m + \widetilde{O} \left( \frac{\overline{\mathcal{G}}_m(\mathcal{F}) + \sqrt{\log(1/\beta)}}{\sqrt{m}} \right).$

Lemma 4 is a standard bound from learning theory [Bousquet, 2002, Rakhlin et al., 2017] and is proved in Appendix C.7 for the sake of completeness. The second component is given by Lemma 13 in Appendix C.5, which amounts to a classical uniform deviations bound for the empirical process, ensuring that if $n \gg 1$, then $L_{\mathcal{D}}(f) \approx L(f)$ for all $f \in \mathcal{F}$. The final step is the following simple lemma, which ensures that if $\eta$ is not too large, then $L_{\mathcal{D}}(\bar{f}) \approx L_{\mathcal{D}}(f_{\mathsf{ERM}})$:

**Lemma 5.** *Let $\mathcal{F} : \mathcal{X} \to [-1, 1]$ be a bounded function class and let $R : \mathcal{F} \to \mathbb{R}$ be an arbitrary, possibly random, regularizer. Let $f_{\mathsf{ERM}} \in \arg\min_{f \in \mathcal{F}} L_{\mathcal{D}}(f)$ and $\bar{f} \in \arg\min_{f \in \mathcal{F}} L_{\mathcal{D}}(f) + R(f)$. Then, $L_{\mathcal{D}}(\bar{f}) \leq L_{\mathcal{D}}(f_{\mathsf{ERM}}) + \sup_{f, f' \in \mathcal{F}} R(f) - R(f')$.*

Lemma 5 is a simple computation proved in Appendix C.5. Letting $R(f)$ be either $\eta \cdot \omega_m^{(j)}(f)$ in Algorithm 2 or $\eta \cdot \|f\|_m^2$ in Algorithm 3 demonstrates that if $\eta$ is not too large, then $\bar{f}$ performs similarly to $f_{\mathsf{ERM}}$. To prove that Algorithms 2 and 3 produce good learning algorithms then, it suffices to combine these three components, observing first that $L(\bar{f})$ is close to optimal if $n \gg 1$ and $\eta$ is not too large, second that $\|\bar{f} - \widehat{f}\|_\mu \ll 1$ if $m \gg 1$ and $\gamma$ is sufficently small, and third that $|L(\widehat{f}) - L(\bar{f})| \lesssim \|\bar{f} - \widehat{f}\|_\mu$ if $\nu_x$ is $\sigma$-smooth with respect to $\mu$ and $\ell$ is $\lambda$-Lipschitz in its first argument. Combining these results concludes the proofs of Theorems 2 and 3. A detailed and rigorous argument for both proofs is presented in Appendix C. As a final remark, we note that in the case of Algorithm 2, convexity of $\ell$ in the first argument is irrelevant to the privacy guarantee despite being necessary for learning. Indeed, for $\bar{f}$ returned by Line 5 in Algorithm 2 to be proven a good learner, we apply Jensen's and the above argument that ensures that $\bar{f}_j$ is a good learner. Interestingly, on the other hand, convexity in $\ell$ is irrelevant to the learning guarantee of Algorithm 3 while it is essential to the privacy guarantee. Further understanding the role that such structural assumptions play in allowing privacy is an interesting direction for future work.

## Acknowledgments and Disclosure of Funding

AB acknowledges support from the National Science Foundation Graduate Research Fellowship under Grant No.1122374 as well as the Simons Foundation and the National Science Foundation through awards DMS-2031883 and DMS-1953181. MB acknowledges support from the National Science Foundation through award NSF CNS-2046425 and from a Sloan Research Fellowship. RD acknowledges support from the National Science Foundation through award NSF CNS-2046425. AS acknowledges support from the Apple AI+ML fellowship. AB also would like to thank Satyen Kale and Claudio Gentile for helpful discussions. Z.S.W. was in part supported by NSF Awards #1763786 and #2339775.

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

## Contents

## A Differentially Private Classification

In the previous section, we presented a private algorithm for general, real-valued loss functions. Here, we turn to the special case of classification, where we provide an algorithm with improved

rates. Formally, binary classification is a special case of Definition 2, where $\mathcal{F} : \mathcal{X} \to \{0, 1\}$ and $\ell$ is the indicator loss.

Much like Algorithms 2 and 3, our approach to classification in Algorithm 4 relies on minimizing a perturbed empirical loss over $\mathcal{F}$ and projecting the output $\widetilde{f}$ onto the public data. Unlike in these earlier algorithms, which require a further perturbation of the output in order to boost stability into differential privacy, in the special case of classification we are able to circumvent this second perturbation and return any $\hat{f}$ that agrees with $\widetilde{f}$ on the public data. This is accomplished by carefully choosing the initial perturbation to the ERM objective (see (4)) so that the predictions of $\widetilde{f}$ on the public data satisfy differential privacy without ensuring some form of stability in norm. As a result, our improved rates then follow from lack of a second perturbation. We present the following guarantee for our classification algorithm, whose pseudo-code can be found in Algorithm 4.

---

**Algorithm 4:** Rounded Report Separator Perturbed Minimum Algorithm (RRSPM)

---

**Input** ERM oracle ERM, dataset $\mathcal{D} = \{(X_i, Y_i) \mid 1 \le i \le n\}$, hypothesis class $\mathcal{F}$, smoothness parameter $\sigma$, loss function $\ell : \mathcal{Y} \times \mathcal{Y} \to \{0, 1\}$, arbitrary $\mathcal{Q} \in \Delta(\mathbb{R})$.
**Draw** $\tilde{\mathcal{D}} = (\tilde{\mathcal{D}}_x, \tilde{\mathcal{D}}_y)$ where $\tilde{\mathcal{D}}_x = \{Z_1, \ldots, Z_m\}$ and $\tilde{\mathcal{D}}_y = \{\tilde{Y}_1, \ldots, \tilde{Y}_m\}$ such that $Z_i \sim \mu$ and $\tilde{Y}_i \sim \text{Uni}(\{0, 1\})$, for all $i \in [m]$.
**Draw** weights $\boldsymbol{\xi} = \{\xi_1, \ldots, \xi_m\}$ such that $\xi_i \sim \text{Lap}(2m/\varepsilon)$.
**Define** $\mathcal{L}_{\boldsymbol{\xi}, \mathcal{D}, \tilde{\mathcal{D}}} : \mathcal{F} \to \mathbb{R}$ such that

$$\mathcal{L}_{\boldsymbol{\xi}, \mathcal{D}, \tilde{\mathcal{D}}}(f) = \sum_{i=1}^{n} \ell(f(X_i), Y_i) + \sum_{i=1}^{m} \xi_i \cdot \ell(f(Z_i), \tilde{Y}_i). \tag{4}$$

**Get** $\tilde{f} = \text{ERM}(\mathcal{F}, \mathcal{L}_{\boldsymbol{\xi}, \mathcal{D}, \tilde{\mathcal{D}}})$.
**Output** $\hat{f} = \text{Perturb}(\tilde{f}, \mathcal{Q}, \gamma = 0, \tilde{D}_x)$                     ▷ By running Algorithm 1

---

**Theorem 4.** *Suppose that $\mathcal{F} : \mathcal{X} \to \mathcal{Y}$ is a function class of VC dimension $d$ and $\ell : \mathcal{Y} \times \mathcal{Y} \to \{0, 1\}$ is the indicator loss. Suppose that $Z_1, \ldots, Z_m \sim \mu$ and $\xi_1, \ldots, \xi_m \sim \text{Lap}(2m/\varepsilon)$ are independent. Then there is a choice of $m$ polynomial in the problem parameters such that if*

$$n = \tilde{\Omega}(d^2 \varepsilon^{-1} \alpha^{-5} \log(\beta^{-1})),$$

*then the $\hat{f}$ returned by Algorithm 4 is $\varepsilon$-pure differentially private. If $\nu_x$ is $\sigma$-smooth with respect to $\mu$, then $\hat{f}$ is an $(\alpha, \beta)$-learner with respect to $\nu_x$ and $\ell$. Furthermore, for some $C > 0$, if $\xi_1, \ldots, \xi_m \sim \mathcal{N}(0, C\sqrt{m \log(1/\delta)}/\varepsilon)$ then there is a choice of $m$ polynomial in the problem parameters such that if*

$$n = \tilde{\Omega}(d^2 \varepsilon^{-1} \alpha^{-4} \log^{1/2}(1/\delta) \log(\beta^{-1})),$$

*then the $\hat{f}$ returned by Algorithm 4 with Gaussian perturbations is $(\varepsilon, \delta)$-differentially private and is an $(\alpha, \beta)$-PAC learner with respect to any $\sigma$-smooth $\nu_x$.*

**Remark 1.** Note that the sample complexity we get in the above theorems is in the general, agnostic setting. In the realizable setting, where some $f^\star \in \mathcal{F}$ perfectly predicts the $Y$ from the $X$, we get a sample complexity of $n = \tilde{\Omega}(d^2 \varepsilon^{-1} \alpha^{-3} \log(\beta^{-1}))$ for $\varepsilon$-pure differential privacy and $n = \tilde{\Omega}(d^2 \varepsilon^{-1} \alpha^{-2.5} \log^{1/2}(1/\delta) \log(\beta^{-1}))$ for $(\varepsilon, \delta)$-differential privacy.

We emphasize that Theorem 4 attains the improved $O(\alpha^{-5})$ sample complexity (even $O(\alpha^{-4}$ for approximate differential privacy)), which is significantly better than the $O(\alpha^{-14})$ from Theorem 2. While this is a major improvement, it still falls short of the desired $O(\alpha^{-2})$ statistical rates achievable by inefficient algorithms from [Bassily et al., 2019]. We leave the interesting question of whether improved sample complexity is possible to future work. We now briefly sketch the proof of Theorem 4.

## A.1 Privacy Analysis of Algorithm 4

While the privacy of Algorithms 2 and 3 is proven in two steps, by first demonstrating stability and then leveraging the output perturbation to ensure privacy, the privacy of Algorithm 4 is proven

directly. Our approach is motivated by techniques from Neel et al. [2019], which adapt the earlier notion of *separator sets* from Goldman et al. [1993], Syrgkanis et al. [2016], Dudík et al. [2020] to the setting of differential privacy. Unlike those works, however, we do not require the strong assumption that $\mathcal{F}$ has a small separator set and our results hold for general VC function classes. The main technical result that ensures privacy of Algorithm 4 demonstrates that the *projection* $\mathcal{F}$ to the public data set $\tilde{\mathcal{D}}_x$ is private with respect to $\mathcal{D}$, where we let $\mathcal{F}|_{\tilde{\mathcal{D}}_x} = \{(f(Z_i))_{1 \leq i \leq m} | f \in \mathcal{F}\}$. We have the following privacy guarantee for $\widetilde{f}(\tilde{D}_x) \in \mathcal{F}|_{\tilde{\mathcal{D}}_x}$:

**Lemma 6.** *(Privacy over Projection) Let $\mathcal{D}, \mathcal{D}'$ be arbitrary datasets containing $n$ points each. Let $\tilde{\mathcal{D}}_x$ be a set of $m$ points $Z_1, \ldots, Z_m \in \mathcal{X}$. Let $\tilde{Y}_1, \ldots, \tilde{Y}_m \in \{0,1\}$ be the set of corresponding labels. Then for all measurable $\mathcal{H} \subseteq \mathcal{F}|_{\tilde{\mathcal{D}}_x}$*

$$\mathbb{P}(\mathsf{ERM}(\mathcal{F}|_{\tilde{\mathcal{D}}_x}, \mathcal{L}_{\boldsymbol{\xi}, \mathcal{D}, \tilde{\mathcal{D}}}) \in \mathcal{H}) \leq e^{\varepsilon} \cdot \mathbb{P}(\mathsf{ERM}(\mathcal{F}|_{\tilde{\mathcal{D}}_x}, \mathcal{L}_{\boldsymbol{\xi}, \mathcal{D}', \tilde{\mathcal{D}}}) \in \mathcal{H}),$$

*where $\mathcal{L}_{\boldsymbol{\xi}, \mathcal{D}, \tilde{\mathcal{D}}}$ is defined as in (4).*

With the above lemma in hand, the privacy of Algorithm 4 follows immediately from the post-processing property of differential privacy. We provide a full proof of Lemma 6 in Appendix D and now turn to the accuracy guarantee.

## A.2 Concluding the Proof of Theorem 4

Proving that Algorithm 4 is an $(\alpha, \beta)$-learner whenever $\nu_x$ is $\sigma$-smooth with respect to $\mu$ is similar to the approach taken in Section 4.3, with the critical difference that in the absence of the second perturbation, we are able to achieve a stronger guarantee on the difference between $\widetilde{f}$ and $\widehat{f}$. Indeed, much as in the previous analysis, we observe that as $m$ increases, the suboptimality of the intermediate $\widetilde{f}$ is driven up, while the difference between $\widetilde{f}$ and $\widehat{f}$ is driven down, thereby requiring a careful balance; here, however, we do not also need to account for the balancing of the variance $\gamma$. We provide a full proof of the accuracy guarantee in Appendix D.

# B  Gaussian Anti-Concentration and Proof of Lemma 1

In this section we present and prove a more general version of Proposition 1. We begin by defining a Gaussian process and then state and prove the result. We then show how Proposition 1 follows as an immediate corollary. To begin, we recall the formal definition of a Gaussian process.

**Definition 7.** Let $T$ be an index set and $m : T \to \mathbb{R}$ be a function. Let $K : T \times T \to \mathbb{R}$ be a covariance kernel in the sense that for any $t_1, \ldots, t_n \in T$, the matrix $(K(t_i, t_j))_{i,j \in [n]}$ is positive semi-definite. We say that $\omega : T \to \mathbb{R}$ is a Gaussian process with mean function $m$ and covariance kernel $K$ if for any $t_1, \ldots, t_n \in T$, the random vector $(\omega(t_i))_{i \in [n]}$ is Gaussian with mean $(m(t_i))_{i \in [n]}$ and covariance matrix $(K(t_i, t_j))_{i,j \in [n]}$. We say that $\omega$ is a centered Gaussian process if $m$ is identically zero.

Note that by Le Gall [2016, Theorem 1.11] such a process always exists given $m, K$. Furthermore, we note that $K$ induces a semi-metric $d$ on $T$ by letting $d(t, t')^2 = \mathbb{E}\left[(\omega(t) - \omega(t'))^2\right]$. We now prove the following result, which is a tighter version of Block et al. [2022, Lemma 33].

**Theorem 5** (Gaussian Anti-concentration). *Let $T$ be a set, $m : T \to \mathbb{R}$ be a mean function and $K : T \times T \to \mathbb{R}$ be a covariance kernel (in the sense of being positive definite). Let $d$ denote the metric induced by $K$ and suppose that $m$ is continuous with respect to $d$, and the metric space $(T, d)$ is separable and compact. Let $\omega$ denote a Gaussian process on $T$ with covariance $K$ and for $\eta > 0$, let*

$$\Omega(t) = m(t) + \eta \cdot \omega(t)$$

*be an offset Gaussian process. We further suppose that $\omega$ is taken to be a version with almost surely continuous paths $t \mapsto \omega(t)$ and that $0 < \kappa \leq K(t, t) \leq 1$ for all $t \in T$. Let*

$$t^{\star} = \operatorname*{argmin}_{t \in T} \Omega(t),$$

*and,*

$$\mathcal{E}(\rho, \tau) = \left\{ \text{there exists } s \in T \text{ such that } \frac{K(s, t^\star)}{K(t^\star, t^\star)} \leq 1 - \rho^2 \text{ and } \Omega(s) \leq \Omega(t^\star) + \tau \right\},$$

*for $\rho, \tau > 0$. The following holds:*

$$\mathbb{P}\left(\mathcal{E}(\rho, \tau)\right) \leq \frac{\tau}{\rho^2 \eta \kappa^2} \cdot \mathbb{E}\left[\sup_{t \in T} \omega(t)\right].$$

Note that $\frac{K(s,t)}{K(t,t)}$ is a measure of how close $s$ and $t$ are to each other; indeed, in the special case where $\kappa = 1$ this is precisely the correlation and thus $s$ and $t$ are more closely related the closer this quantity is to 1. Thus the event $\mathcal{E}(\rho, \tau)$ can be interpreted to mean that there exists some point $s$ far from $t^\star$ (as governed by $\rho$) such that $\Omega(s)$ is almost minimal (as governed by $\tau$); in other words, Theorem 5 puts an upper bound on the probability that almost-minimizers of a Gaussian process lie far from the true minimizer. We now prove Theorem 5.

*Proof of Theorem 5.* Note that by compactness of $T$ and almost sure continuity of $\Omega$, a minimizer of $\Omega$ exists almost surely; furthermore, by [Kim and Pollard, 1990, Lemma 2.6], $t^\star$ is almost surely unique. As $T$ is separable and $\Omega$ has almost surely continuous sample paths, it suffices to replace $T$ with a countable dense subset. We will hereafter suppose without loss of generality that $T$ is countable. For each $t \in T$, define the set

$$A(t) = \left\{ s \in T \,\middle|\, \frac{K(s, t)}{K(t, t)} \leq 1 - \rho^2 \text{ and } \Omega(s) \leq \Omega(t) + \tau \right\}.$$

It then suffices to lower bound the probability that $A(t^\star) = \emptyset$. We compute

$$\mathbb{P}\left(|A(t^\star)| = 0\right) = \sum_{t \in T} \mathbb{P}\left(t^\star = t \text{ and } |A(t)| = 0\right)$$

$$= \sum_{t \in T} \mathbb{E}_y\left[\mathbb{P}\left(t^\star = t \text{ and } \inf_{K(s,t) \leq (1-\rho^2)K(t,t)} \Omega(s) \geq y + \tau \,\middle|\, \Omega(t) = y\right)\right],$$

where the expectation is taken over the distribution of $\Omega(t)$. Now, fix $t$ and let $\Omega_{t,y}$ denote the Gaussian process $\Omega$ conditioned on the event that $\Omega(t) = y$. Let $m_{t,y}$ and $\eta^2 \cdot K_t$ denote the mean and covariance processes of $\Omega_{t,y}$. Critically, note that $K_t$ is independent of $y$ and for all $s \neq t$, we have

$$m_{t,y}(s) = m(s) + \frac{K(s, t)}{K(t, t)}(y - m(t)).$$

Define the functions

$$a(s) = \frac{\tau}{\rho^2} \cdot \frac{K(s, t)}{K(t, t)} \qquad \text{and} \qquad b(s) = \frac{\tau}{\rho^2} - a(s).$$

Now, note that if $K(s, t) \leq (1 - \rho^2) \cdot K(t, t)$, then

$$b(s) = \frac{\tau}{\rho^2}\left(1 - \frac{K(s, t)}{K(t, t)}\right) \geq \frac{\tau}{\rho^2} \cdot \rho^2 = \tau. \tag{5}$$

We also have that $b(s) \geq 0$ for all $s$ by the fact that $K(s, t) \leq \sqrt{K(s, s) \cdot K(t, t)}$ and $K(s, s) \vee K(t, t) \leq 1$. Furthermore, for all $s$, it holds that

$$m_{t, y + \frac{\tau}{\rho^2}}(s) = m_{t,y}(s) + a(s). \tag{6}$$

Thus, for fixed $t \in T$ and $y \in \mathbb{R}$, we have

$$\mathbb{P}\left(t^\star = t \text{ and } \inf_{K(s,t) \leq (1-\rho^2)K(t,t)} \Omega(s) \geq y + \tau | \Omega(t) = y\right)$$

$$\geq \mathbb{P}\left(t^\star = t \text{ and } \inf_{K(s,t) \leq (1-\rho^2)K(t,t)} \Omega(s) - b(s) \geq y | \Omega(t) = y\right)$$

$$= \mathbb{P}\left(t^\star = t \text{ and } \inf_{K(s,t) \leq (1-\rho^2)K(t,t)} \Omega(s) - b(s) - a(s) + a(s) \geq y | \Omega(t) = y\right)$$

$$= \mathbb{P}\left(t^\star = t \text{ and } \inf_{K(s,t) \leq (1-\rho^2)K(t,t)} \Omega(s) + a(s) \geq y + \frac{\tau}{\rho^2} | \Omega(t) = y\right)$$

$$= \mathbb{P}\left(t^\star = t \text{ and } \inf_{K(s,t) \leq (1-\rho^2)K(t,t)} \Omega(s) \geq y + \frac{\tau}{\rho^2} | \Omega(t) = y + \frac{\tau}{\rho^2}\right)$$

$$\geq \mathbb{P}\left(t^\star = t \text{ and } \inf_{K(s,t) \leq (1-\rho^2)K(t,t)} \Omega(s) \geq y + \frac{\tau}{\rho^2} | \Omega(t) = y + \frac{\tau}{\rho^2}\right),$$

where the first inequality follows from (5), the second equality follows from the construction, and the last equality follows from (6) and the fact that $K_t$ is independent of $y$. Now, denote

$$q_t(y) = (2\pi K(t,t))^{-\frac{1}{2}} \exp\left(-\frac{(y - m(t))^2}{2\eta^2 K(t,t)}\right),$$

the density of $\Omega(t)$ and note that we have

$$\mathbb{P}\left(|A(t^\star)| = 0\right) = \sum_{t \in T} \int_{-\infty}^{\infty} q_t(y) \mathbb{P}\left(t^\star = t \text{ and } \inf_{K(s,t) \leq 1-\rho^2} \Omega(s) \geq y + \tau | \Omega(t) = y\right) dy$$

$$\geq \sum_{t \in T} \int_{\infty}^{\infty} q_t(y) \mathbb{P}\left(t^\star = t \text{ and } \inf_{K(s,t) \leq 1-\rho^2} \Omega(s) \geq y + \frac{\tau}{\rho^2} | \Omega(t) = y + \frac{\tau}{\rho^2}\right) dy. \tag{7}$$

We then compute

$$\int_{-\infty}^{\infty} q_t(y) \mathbb{P}\left(t^\star = t \text{ and } \inf_{K(s,t) \leq 1-\rho^2} \Omega(s) \geq y + \frac{\tau}{\rho^2} | \Omega(t) = y + \frac{\tau}{\rho^2}\right) dy \tag{8}$$

$$= \int_{-\infty}^{\infty} q_t(y) \mathbb{P}\left(t^\star = t \text{ and } \inf_{K(s,t) \leq 1-\rho^2} \Omega(s) \geq y | \Omega(t) = y\right) dy$$

$$+ \int_{-\infty}^{\infty} \left(q_t(y) - q_t\left(y - \frac{\tau}{\rho^2}\right)\right) \cdot \mathbb{P}\left(t^\star = t \text{ and } \inf_{K(s,t) \leq 1-\rho^2} \Omega(s) \geq y | \Omega(t) = y\right) dy,$$

where we added and subtracted the first term and then made the variable substitution $y + \frac{\tau}{\rho^2} \mapsto y$ for the latter integral. Note that

$$\mathbb{P}\left(t^\star = t \text{ and } \inf_{K(s,t) \leq 1-\rho^2} \Omega(s) \geq y | \Omega(t) = y\right) = \mathbb{P}\left(t^\star = t | \Omega(t) = y\right)$$

as $\Omega(s) \geq \Omega(t^\star)$ for all $s \in T$ by definition. Combining this observation with (7) and (8) yields

$$\mathbb{P}\left(|A(t^\star)| = 0\right) \geq \sum_{t \in T} \int_{-\infty}^{\infty} q_t(y) \mathbb{P}\left(t^\star = t | \Omega(t) = y\right) dy$$

$$- \sum_{t \in T} \int_{-\infty}^{\infty} \left(q_t(y) - q_t\left(y - \frac{\tau}{\rho^2}\right)\right) \cdot \mathbb{P}\left(t^\star = t | \Omega(t) = y\right) dy.$$

For the first term, we have

$$\sum_{t \in T} \int_{-\infty}^{\infty} q_t(y) \mathbb{P}\left(t^\star = t | \Omega(t) = y\right) dy = \sum_{t \in T} \mathbb{P}\left(t^\star = t\right) = 1.$$

For the second term, using the fact that $1 - e^x \leq x$ for all $x$, we have

$$q_t(y) - q_t\left(y - \frac{\tau}{\rho^2}\right) = q_t(y)\left(1 - \exp\left(\frac{(y-m(t))^2}{2\eta^2 K(t,t)} - \frac{\left(y - m(t) - \frac{\tau}{\rho^2}\right)^2}{2\eta^2 K(t,t)}\right)\right)$$

$$\leq q_t(y)\left(\frac{(y-m(t))^2}{2\eta^2 K(t,t)} - \frac{\left(y - m(t) - \frac{\tau}{\rho^2}\right)^2}{2\eta^2 K(t,t)}\right)$$

$$\leq \frac{q_t(y)}{2\eta^2 \kappa^2} \cdot \left(\frac{2\tau}{\rho^2}(y - m(t))\right).$$

Thus we have

$$\mathbb{P}\left(|A(t^\star)| > 0\right) = 1 - \mathbb{P}\left(|A(t^\star)| = 0\right)$$

$$\leq \sum_{t \in T} \int_{-\infty}^{\infty} \frac{q_t(y)}{2\eta^2 \kappa^2} \cdot \left(\frac{2\tau}{\rho^2}(y - m(t))\right) \cdot \mathbb{P}\left(t^\star = t | \Omega(t) = y\right) dy$$

$$= \frac{\tau}{\rho^2 \eta^2 \kappa^2} \sum_{t \in T} \int_{-\infty}^{\infty} (y - m(t)) q_t(y) \mathbb{P}\left(t^\star = t | \Omega(t) = y\right) dy$$

$$= \frac{\tau}{\rho^2 \eta^2 \kappa^2} \cdot \mathbb{E}\left[\Omega(t^\star) - m(t^\star)\right]$$

$$\leq \frac{\tau}{\rho^2 \eta^2 \kappa^2} \cdot \mathbb{E}\left[\sup_{t \in T} \Omega(t) - m(t)\right].$$

The result follows by noting that $\eta \cdot \omega(t) = \Omega(t) - m(t)$ for all $t \in T$. $\qquad \square$

We now prove a corollary of [Theorem 5](#) that will be useful in the proof of [Proposition 1](#) and which makes the relationship between $\mathcal{E}(\rho, \tau)$ and the intuition of distance between $t^\star$ and $s$ more explicit.

**Corollary 1.** *Suppose that we are in the situation of [Theorem 5](#) with the additional conditions that $T$ is a subset of a real vector space and that $d(s,t) = \sqrt{K(s-t, s-t)}$. Let $m' : T \to \mathbb{R}$ denote a mean function such that $\sup_{t \in T} |m(t) - m'(t)| \leq \tau$ and let $\Omega'$ denote the corresponding shifted Gaussian process. If $t^{\star'} = \operatorname{argmin}_{t \in T} \Omega'(t)$, then*

$$\mathbb{P}\left(d(t^\star, t^{\star'}) > \rho\right) \leq \frac{8\tau}{\rho^4 \eta \kappa^2} \cdot \mathbb{E}\left[\sup_{t \in T} \omega(t)\right].$$

*Proof.* Note that

$$d(s,t)^2 = K(s-t, s-t) = K(s,s) + K(t,t) - 2K(s,t).$$

Let $M = \max(K(t^\star, t^\star), K(t^{\star'}, t^{\star'})) \leq 1$ and note that the above implies:

$$d(t^\star, t^{\star'})^2 \leq 2M\left(1 - \frac{K(t^{\star'}, t^\star)}{M}\right).$$

Thus,

$$\mathbb{P}\left(d(t^\star, t^{\star'}) > \rho\right) \leq \mathbb{P}\left(2M\left(1 - \frac{K(t^\star, t^{\star'})}{M}\right) > \rho\right)$$

$$\leq \mathbb{P}\left(1 - \frac{K(t^\star, t^{\star'})}{M} > \frac{\rho^2}{2}\right)$$

$$\leq 2\left(\frac{\tau}{\left(\frac{\rho^2}{2}\right)^2 \eta \kappa^2} \cdot \mathbb{E}\left[\sup_{t \in T} \omega(t)\right]\right)$$

$$= \frac{8\tau}{\rho^4 \eta \kappa^2} \cdot \mathbb{E}\left[\sup_{t \in T} \omega(t)\right],$$

where the last inequality follows by applying a union bound and Theorem 5 to the Gaussian processes $\Omega$ and $\Omega'$ after observing that

$$\Omega'(t^\star) \leq \Omega'(t^{\star'}) + \tau$$

and similarly for $\Omega(t^{\star'})$. $\qquad\square$

We are now ready to prove Proposition 1 using Corollary 1.

*Proof of Proposition 1.* Let $T = \mathcal{F}$, and observe that

$$d(f, f')^2 = K(f - f', f - f') = \mathbb{E}\left[\omega(f) - \omega(f')^2\right] = \|f - f'\|^2.$$

The result then follows immediately by applying Corollary 1 to the Gaussian process $\Omega(f) = m(f) + \eta \cdot \omega(f)$. $\qquad\square$

## C  Analysis of Algorithms 2 and 3

In this section we provide the full proofs for Theorems 2 and 3, as well as more general statements under different measures of the complexity of the function class $\mathcal{F}$. We begin the section by stating formal bounds on the learning and differential privacy guarantees for each algorithm. In Appendix C.1, we prove Lemma 1, which is a key technical lemma in the differential privacy guarantee of Algorithm 2; we then continue in Appendix C.2 by proving Lemma 2, which plays an analogous role except in the analysis of Algorithm 3. In Appendix C.3, we prove that algorithmic stability in $\|\cdot\|_m$ can be boosted to differential privacy through Algorithm 1. In Appendix C.4 we combine the previous results to give guarantees on the differential privacy of Algorithm 2 and Algorithm 3. We continue in Appendix C.5 by applying more standard learning theoretic techniques to demonstrate that both algorithms are PAC learners before concluding the proofs of the main theorems in Appendix C.6. Finally, for the sake of completeness, we prove a norm comparison lemma in Appendix C.7 that was deferred from Appendix C.5 for the purpose of continuity.

### C.1  Stability of Algorithm 2 and Proof of Lemma 1

We break the proof into two parts. First, we integrate the tail bound from Proposition 1 to get control on $\mathbb{E}\left[\left\|\bar{f}_j - \bar{f}'_j\right\|_m\right]$ for any $j \in [J]$. We then apply a Jensen's inequality and a Chernoff bound to get high probability control on $\left\|\bar{f} - \bar{f}'\right\|$. We begin with the following lemma:

**Lemma 7.** *Let $\bar{f}_j$ be as in in Line 6 of Algorithm 2 and $\bar{f}'_j$ is defined analogously with respect to $\mathcal{D}'$, a neighboring dataset to $\mathcal{D}$. Then*

$$\mathbb{E}\left[\left\|\bar{f}_j - \bar{f}'_j\right\|_m^2\right] \leq \frac{2}{(n \cdot \eta)^{1/3}\kappa^{2/3}} \cdot \left(\mathbb{E}\left[\sup_{f \in \mathcal{F}} \omega_m(f)\right]\right)^{1/3},$$

*where the expectation is with respect to $\xi^{(j)}$.*

*Proof.* By boundedness of $\ell$, it holds that $|L_\mathcal{D}(f) - L_{\mathcal{D}'}(f)| \leq \frac{1}{n}$ for all $f \in \mathcal{F}$. Thus, if $\bar{f}^{(1)}$ and $\bar{f}^{(1)'}$ are as in the statement of the corollary, then $L_\mathcal{D}(\bar{f}^{(1)'}) + \eta \cdot \omega_m^{(1)}(\bar{f}^{(1)'}) \leq L_\mathcal{D}(\bar{f}^{(1)}) + \eta \cdot \omega_m^{(1)}(\bar{f}^{(1)}) + \frac{1}{n}$. Plugging in $\tau = \frac{1}{n}$ and applying Proposition 1 yields

$$\mathbb{P}\left(\left\|\bar{f}_j - \bar{f}'_j\right\|_m > \rho\right) \leq \frac{8}{n\rho^4\kappa^2\eta} \cdot \mathbb{E}\left[\sup_{f \in \mathcal{F}} \omega_m(f)\right].$$

Now, we can integrate the tail bound to get for any $\zeta > 0$,

$$
\mathbb{E}\left[\left\|\bar{f} - \bar{f}'\right\|_m^2\right] \leq \zeta + \int_\zeta^1 2\rho \mathbb{P}\left(\left\|\bar{f} - \bar{f}'\right\|_m > \rho\right) d\rho
$$

$$
\leq \zeta + 2 \cdot \int_\zeta^1 \rho \cdot \frac{8}{n\rho^4\kappa^2\eta} \cdot \mathbb{E}\left[\sup_{f \in \mathcal{F}} \omega_m(f)\right] d\rho
$$

$$
\leq \zeta + \frac{8}{n\zeta^2\kappa^2\eta} \cdot \mathbb{E}\left[\sup_{f \in \mathcal{F}} \omega_m(f)\right] \cdot \log\left(\frac{1}{\zeta}\right).
$$

Minimizing over $\zeta$ yields the result. $\qquad\square$

We now apply a Jensen's inequality and a Chernoff bound to get high probability control on $\left\|\bar{f} - \bar{f}'\right\|$.

*Proof of Lemma 1.* By Jensen's inequality and the definition of $\bar{f}$, it holds that

$$
\mathbb{P}\left(\left\|\bar{f} - \bar{f}'\right\|_m > \rho\right) = \mathbb{P}\left(\left\|\frac{1}{J}\sum_{j=1}^J \bar{f}_j - \bar{f}'_j\right\|_m > \rho\right)
$$

$$
= \mathbb{P}\left(\left\|\frac{1}{J}\sum_{j=1}^J \bar{f}_j - \bar{f}'_j\right\|_m^2 > \rho^2\right)
$$

$$
\leq \mathbb{P}\left(\frac{1}{J}\sum_{j=1}^J \left\|\bar{f}_j - \bar{f}'_j\right\|_m^2 > \rho^2\right).
$$

Finally, by Hoeffding's inequality [Wainwright, 2019, Van Handel, 2014] and Lemma 7, we have that

$$
\delta \geq \mathbb{P}\left(\frac{1}{J}\sum_{j=1}^J \left\|\bar{f}_j - \bar{f}'_j\right\|_m^2 > \mathbb{E}\left[\left\|\bar{f}_1 - \bar{f}'_1\right\|_m\right] + \sqrt{\frac{\log\left(\frac{1}{\delta}\right)}{J}}\right)
$$

$$
\geq \mathbb{P}\left(\frac{1}{J}\sum_{j=1}^J \left\|\bar{f}_j - \bar{f}'_j\right\|_m^2 > \frac{2}{(n \cdot \eta)^{1/3}\kappa^{2/3}} \cdot \left(\mathbb{E}\left[\sup_{f \in \mathcal{F}} \omega_m(f)\right]\right)^{1/3} + \sqrt{\frac{\log\left(\frac{1}{\delta}\right)}{J}}\right).
$$

The result follows. $\qquad\square$

## C.2   Stability of Algorithm 3 and Proof of Lemma 2

In this section we prove Lemma 2 based on a technique borrowed from online learning and the analysis of Follow the Regularized Leader (FTRL) [Gordon, 1999, Cesa-Bianchi and Lugosi, 2006].

*Proof of Lemma 2.* Let $\mathcal{L}$ be as in (3) and $\mathcal{L}'$ be defined similarly but with $\mathcal{D}$ replaced by $\mathcal{D}'$. Note that $\|\cdot\|_m^2$ is strongly convex with respect to $\|\cdot\|_m$ [Rockafellar, 2015]. Thus, by convexity of $\mathcal{F}$, it holds that

$$
\mathcal{L}(\bar{f}') \geq \mathcal{L}(\bar{f}) + \frac{\eta}{2}\left\|\bar{f} - \bar{f}'\right\|_m^2.
$$

On the other hand,

$$
\mathcal{L}(\bar{f}') = \mathcal{L}'(\bar{f}') + \mathcal{L}(\bar{f}') - \mathcal{L}'(\bar{f}')
$$

$$
\leq \mathcal{L}'(\bar{f}) + \mathcal{L}(\bar{f}') - \mathcal{L}'(\bar{f}')
$$

$$
= \mathcal{L}(\bar{f}) + \mathcal{L}(\bar{f}') - \mathcal{L}'(\bar{f}') + \mathcal{L}'(\bar{f}) - \mathcal{L}(\bar{f})
$$

$$
\leq \mathcal{L}(\bar{f}) + \frac{2}{n}.
$$

Combining this with the previous display and rearranging yields:

$$\eta \cdot \left\| \bar{f} - \bar{f}' \right\|_m^2 \leq \frac{4}{n}.$$

Rearranging again proves the result. $\square$

### C.3 Boosting Stability to Differential Privacy: Proofs from Section 4.2

In this section, we analyze Algorithm 1 and prove that if $\bar{f}$ is stable in $\|\cdot\|_m$ then applying the output perturbation yields a differentially private algorithm for standard choices of perturbation distribution $\mathcal{Q}$. We also show that Algorithm 1 returns $\widehat{f}$ close to the $\bar{f}$ with high probability. This first claim is implied by the following standard concentration bound.

**Lemma 8.** *Suppose that $\mathcal{Q} = \mathcal{N}(0,1)$ and let $\widehat{f} = \mathsf{Perturb}(\bar{f}, \mathcal{Q}, \gamma, \widetilde{\mathcal{D}}_x)$ be as in Algorithm 1. Then with probability at least $1 - \beta$,*

$$\left\| \bar{f} - \widehat{f} \right\|_m \leq 2\gamma \cdot \sqrt{\log\left(\frac{1}{\beta}\right)}. \tag{9}$$

*If $\mathcal{Q} = Lap(1)$, then with probability at least $1 - \beta$,*

$$\left\| \bar{f} - \widehat{f} \right\|_m \leq 2\gamma \cdot \log\left(\frac{1}{\beta}\right) \cdot \sqrt{m}. \tag{10}$$

*Proof.* By construction, it holds that

$$\left\| \widehat{f} - \bar{f} - \gamma \cdot \zeta \right\|_m \leq \left\| \bar{f} - \bar{f} - \gamma \cdot \zeta \right\|_m = \gamma \cdot \|\zeta\|_m.$$

By the triangle inequality, it holds that

$$\left\| \widehat{f} - \bar{f} \right\|_m \leq \left\| \widehat{f} - \bar{f} - \gamma \cdot \zeta \right\|_m + \gamma \cdot \|\zeta\|_m \leq 2\gamma \cdot \|\zeta\|_m.$$

Thus it suffices to bound $\|\zeta\|_m$. Bounds on this quantity when $\zeta \sim \mathcal{N}(0,1)$ or $\zeta \sim Lap(1)$ are standard and can be found in, for example Wainwright [2019]. The result follows. $\square$

We now prove the main property of Algorithm 1, namely that it boosts stability to differential privacy.

**Lemma 9.** *Let $\mathsf{Perturb}$ be as in Algorithm 1 and suppose that $\bar{f}, \bar{f}' \in \mathcal{F}$. If $\widetilde{\mathcal{D}}_x = \{Z_1, \ldots, Z_m\} \subset \mathcal{X}$ is arbitrary and $\mathcal{Q} = \mathcal{N}(0,1)$, then for any $\delta > 0$ and any measurable $\mathcal{G} \subset \mathcal{F}$, it holds that*

$$\mathbb{P}\left( \mathsf{Perturb}(\bar{f}, \mathcal{Q}, \gamma, \widetilde{\mathcal{D}}_x) \in \mathcal{G} \right) \leq e^{\frac{m}{2\gamma^2}\left(1 + \gamma \cdot \sqrt{\log\left(\frac{1}{\delta}\right)}\right) \cdot \|\bar{f} - \bar{f}'\|_m} \cdot \mathbb{P}\left( \mathsf{Perturb}(\bar{f}', \mathcal{Q}, \gamma, \widetilde{\mathcal{D}}_x) \in \mathcal{G} \right) + \delta.$$

*On the other hand, if $\mathcal{Q} = Lap(1)$, then*

$$\mathbb{P}\left( \mathsf{Perturb}(\bar{f}, \mathcal{Q}, \gamma, \widetilde{\mathcal{D}}_x) \in \mathcal{G} \right) \leq e^{\frac{m^{3/2}}{\gamma} \cdot \|\bar{f}' - \bar{f}\|_m} \cdot \mathbb{P}\left( \mathsf{Perturb}(\bar{f}', \mathcal{Q}, \gamma, \widetilde{\mathcal{D}}_x) \in \mathcal{G} \right)$$

*Proof.* To prove the first statement, let $\mathcal{B}_\delta$ denote the event that $\|\zeta\|_m \leq \gamma \cdot \sqrt{\log\left(\frac{1}{\delta}\right)}$ and let

$$p(u) = (2\pi\gamma^2)^{-\frac{m}{2}} \cdot \exp\left(-\frac{1}{2\gamma^2} \cdot \|u\|_m^2\right)$$

be the density of $\gamma \cdot \zeta$. Note that $\mathbb{P}(\mathcal{B}_\delta^c) \leq \delta$ by Lemma 8. We compute:

$$\mathbb{P}\left( \mathsf{Perturb}(\bar{f}, \mathcal{Q}, \gamma, \widetilde{\mathcal{D}}_x) \in \mathcal{G} \right) = \int_{u \in \mathbb{R}^m} \mathbb{P}\left( \mathsf{Perturb}(\bar{f}, \mathcal{Q}, \gamma, \widetilde{\mathcal{D}}_x) \in \mathcal{G} | \gamma \cdot \zeta = u \right) p(u) du$$

$$= \int_{u \in \mathbb{R}^m} \mathbb{I}\left[\mathcal{B}_\delta\right] \cdot \mathbb{P}\left( \mathsf{Perturb}(\bar{f}, \mathcal{Q}, \gamma, \widetilde{\mathcal{D}}_x) \in \mathcal{G} | \gamma \cdot \zeta = u \right) p(u) du$$

$$+ \int_{u \in \mathbb{R}^m} \mathbb{I}\left[\mathcal{B}_\delta^c\right] \mathbb{P}\left( \mathsf{Perturb}(\bar{f}, \mathcal{Q}, \gamma, \widetilde{\mathcal{D}}_x) \in \mathcal{G} | \gamma \cdot \zeta = u \right) p(u) du$$

$$\leq \delta + \int_{u \in \mathbb{R}^m} \mathbb{I}\left[\mathcal{B}_\delta\right] \cdot \mathbb{P}\left( \mathsf{Perturb}(\bar{f}, \mathcal{Q}, \gamma, \widetilde{\mathcal{D}}_x) \in \mathcal{G} | \gamma \cdot \zeta = u \right) p(u) du.$$

For the second term, we compute:

$$\int_{u\in\mathbb{R}^m} \mathbb{I}\left[\mathcal{B}_\delta\right] \cdot \mathbb{P}\left(\mathsf{Perturb}(\bar{f}, \mathcal{Q}, \gamma, \widetilde{\mathcal{D}}_x) \in \mathcal{G} | \gamma \cdot \zeta = u\right) p(u) du$$

$$= \int_{u\in\mathbb{R}^m} \mathbb{I}\left[\mathcal{B}_\delta\right] \cdot \mathbb{P}\left(\mathsf{Perturb}(\bar{f}', \mathcal{Q}, \gamma, \widetilde{\mathcal{D}}_x) \in \mathcal{G} | \gamma \cdot \zeta = u + \bar{f}' - \bar{f}\right) p(u) du$$

$$= \int_{u\in\mathbb{R}^m} \mathbb{I}\left[\mathcal{B}_\delta\right] \cdot \mathbb{P}\left(\mathsf{Perturb}(\bar{f}', \mathcal{Q}, \gamma, \widetilde{\mathcal{D}}_x) \in \mathcal{G} | \gamma \cdot \zeta = u\right) p(u) \cdot e^{\frac{m}{2\gamma^2}\left(\|u - \bar{f}'\|_m^2 - \|u - \bar{f}\|_m^2\right)} du$$

$$\leq \int_{u\in\mathbb{R}^m} \mathbb{I}\left[\mathcal{B}_\delta\right] \cdot \mathbb{P}\left(\mathsf{Perturb}(\bar{f}', \mathcal{Q}, \gamma, \widetilde{\mathcal{D}}_x) \in \mathcal{G} | \gamma \cdot \zeta = u\right) p(u) \cdot e^{\frac{m}{2\gamma^2}\left(1 + \|u\|_m\right) \cdot \|\bar{f} - \bar{f}'\|_m} du$$

$$\leq \int_{u\in\mathbb{R}^m} \mathbb{I}\left[\mathcal{B}_\delta\right] \cdot \mathbb{P}\left(\mathsf{Perturb}(\bar{f}', \mathcal{Q}, \gamma, \widetilde{\mathcal{D}}_x) \in \mathcal{G} | \gamma \cdot \zeta = u\right) p(u) \cdot e^{\frac{m}{2\gamma^2}\left(1 + \gamma \cdot \sqrt{\log\left(\frac{1}{\delta}\right)}\right) \cdot \|\bar{f} - \bar{f}'\|_m} du$$

$$\leq e^{\frac{m}{2\gamma^2}\left(1 + \gamma \cdot \sqrt{\log\left(\frac{1}{\delta}\right)}\right) \cdot \|\bar{f} - \bar{f}'\|_m} \cdot \mathbb{P}\left(\mathsf{Perturb}(\bar{f}', \mathcal{Q}, \gamma, \widetilde{\mathcal{D}}_x) \in \mathcal{G}\right).$$

The first claim follows. To prove the second claim, we may repeat the same argument with $\mathcal{Q} = Lap(1)$ and $\delta = 0$. Indeed, observe that if

$$q(u) = \gamma^{-m} \cdot e^{-\frac{\|u\|_{\ell 1}}{\gamma}}$$

then

$$\frac{q(u + \bar{f}' - \bar{f})}{q(u)} \leq e^{\frac{\|\bar{f}' - \bar{f}\|_{\ell 1}}{\gamma}} \leq e^{\frac{m^{3/2}}{\gamma} \cdot \|\bar{f}' - \bar{f}\|_m},$$

where the second inequality follows from Cauchy-Schwarz. Plugging this ratio into the above argument yields the second claim. $\qquad\square$

## C.4 Concluding the Proofs of Differential Privacy

In this section, we combine the results from Appendix C.3 with the stability results from Appendix C.6 to prove the differential privacy guarantees of Algorithm 2 and Algorithm 3. We separate this section into two lemmas, each corresponding to one of the algorithms. We begin with the more general result. Note that this lemma does not quite follow immediately from combining the stability guarantee with the results of Appendix C.3 as we wish to assume a uniform lower bound on $\|f\|_\mu$ whereas Lemma 1 requires a uniform lower bound on $\|\cdot\|_m$. We apply a result from Appendix C.5 below to reconcile this discrepancy.

**Lemma 10** (Differential Privacy Guarantee for Algorithm 2). *Suppose that $\mathcal{F}: \mathcal{X} \to [-1, 1]$ is a function class and let $\mu \in \Delta(\mathcal{X})$ such that $\|f\|^2 \geq \frac{2}{3}$ for all $f \in \mathcal{F}$. Suppose further that $\ell$ is bounded in $[0, 1]$. Let $\delta > 0$ and suppose that $m, \gamma, \eta$, and $J$ are such that*

$$\frac{m}{2\gamma^2}\left(1 + \gamma \cdot \sqrt{\log\left(\frac{2}{\delta}\right)}\right)\left(\frac{4}{(n \cdot \eta)^{1/3}} \cdot \mathbb{E}\left[\sup_{f \in \mathcal{F}} \omega_m(f)\right] + \sqrt{\frac{\log\left(\frac{2}{\delta}\right)}{J}}\right) \leq \varepsilon$$

*and*

$$C\left(\frac{\log^2(m)}{\sqrt{m}} \cdot \overline{\mathcal{G}}_m(\mathcal{F}) + \sqrt{\frac{\log\log(m) + \log\left(\frac{1}{\delta}\right)}{m}}\right) \leq \frac{1}{6}.$$

*Then Algorithm 2 is $(\varepsilon, \delta)$-differentially private for $\mathcal{Q} = \mathcal{N}(0, 1)$.*

*Proof.* The result follows immediately by combining Lemmas 1 and 9 assuming we have a lower bound on $\kappa$. Indeed, by Lemma 12, it holds with probability at least $1 - \delta$ that $\inf_{f \in \mathcal{F}} \|f\|_m \geq \frac{1}{4}$. The result follows. $\qquad\square$

We also have a guarantee for the more specialized algorithm.

**Lemma 11** (Differential Privacy Guarantee for Algorithm 3). *Suppose that $\mathcal{F} : \mathcal{X} \to [-1, 1]$ is a convex function class and suppose that $\ell$ is convex and $\lambda$-Lipschitz in its first argument. If $\delta > 0$ and $m, \gamma, \eta$ are such that*

$$\frac{m}{2\gamma^2}\left(1 + \gamma \cdot \sqrt{\log\left(\frac{2}{\delta}\right)}\right) \cdot \frac{2}{\sqrt{\eta \cdot n}} \le \varepsilon,$$

*then Algorithm 3 run with $\mathcal{Q} = \mathcal{N}(0, 1)$ is $(\varepsilon, \delta)$-differentially private. On the other hand, if $\delta = 0$ and*

$$\frac{m^{3/2}}{\gamma} \cdot \frac{2}{\sqrt{\eta \cdot n}} \le \varepsilon,$$

*then Algorithm 3 run with $\mathcal{Q} = Lap(1)$ is $\varepsilon$-purely differentially private.*

*Proof.* This follows immediately by combining Lemmas 2 and 9. $\qquad\square$

These results show that for any choice of $\nu$, Algorithms 2 and 3 are differentially private. In the next section we show that if $\nu$ is $\sigma$-smooth with respect to $\mu$, then the algorithms are also PAC learners with respect to $\nu$.

## C.5 PAC guarantees for Algorithms 2 and 3

The previous sections have shown that Algorithms 2 and 3 are differentially private, which comprises the main difficulty of our analysis. Here we apply standard learning theoretic techniques to show that if $\nu$ is $\sigma$-smooth with respect to $\mu$, then the algorithms are also PAC learners with respect to $\nu$. This proof rests on three main results: first, we recall a norm comparison guarantee in high probability that allows us to relate $\|\cdot\|_\mu$ to $\|\cdot\|_m$; second, we recall a classical uniform deviations bound for empirical processes; and third, we show that the perturbed empirical minimizer $\bar{f}$ has similar loss to the empirical minimizer $f_{\mathsf{ERM}}$ of a loss function as long as the perturbation is not too large. Combining all three results will result in a PAC learning guarantee for Algorithms 2 and 3.

We begin with the following lemma, which is a fairly standard result in learning theory. To state the lemma, we recall from Definition 6 that the worst-case Gaussian complexity is defined as

$$\overline{\mathcal{G}}_m(\mathcal{F}) = \sup_{Z_1, \ldots, Z_m} \mathbb{E}\left[\sup_{f \in \mathcal{F}} \omega_m(f)\right],$$

We then have the following control on $\|\cdot\|_\mu$ in terms of $\|\cdot\|_m$:

**Lemma 12.** *Suppose that $\mathcal{F} : \mathcal{X} \to [-1, 1]$ is a function class and let $\mu \in \Delta(\mathcal{X})$ with $Z_1, \ldots, Z_m \sim \mu$ independent. Then for any $\beta > 0$, it holds with probability at least $1 - \beta$ that for all $f, f' \in \mathcal{F}$,*

$$\|f - f'\|_\mu \le 2 \cdot \|f - f'\|_m + C\left(\frac{\log^2(m)}{\sqrt{m}} \cdot \overline{\mathcal{G}}_m(\mathcal{F}) + \sqrt{\frac{\log\log(m) + \log\left(\frac{1}{\beta}\right)}{m}}\right).$$

Because the proof is relatively standard [Bousquet, 2002, Rakhlin et al., 2017], but also a technical digression, we defer it to Appendix C.7 and continue with our arguments.

Our second lemma is a standard uniform deviation bound for empirical processes:

**Lemma 13.** *Let $\mathcal{F} : \mathcal{X} \to [-1, 1]$ be a bounded function class and let $\mathcal{D}$ denote a data set of $(X_i, Y_i) \sim \nu$ be independent. Then for any $\beta > 0$, with probability at least $1 - \beta$, it holds that*

$$\sup_{f \in \mathcal{F}} |L_{\mathcal{D}}(f) - L(f)| \le \frac{6}{\sqrt{n}} \cdot \mathbb{E}\left[\sup_{f \in \mathcal{F}} \omega_n(f)\right] + \sqrt{\frac{2\log\left(\frac{1}{\beta}\right)}{n}}.$$

*Proof.* This follows immediately from combining Wainwright [2019, Theorem 4.10] with Van Handel [2014, Lemma 7.4]. □

Finally, we show that the perturbed empirical minimizer $\bar{f}$ has similar loss to the empirical minimizer $f_{\mathsf{ERM}}$ of a loss function as long as the perturbation is not too large.

**Lemma 14.** *Let $\mathcal{F} : \mathcal{X} \to [-1,1]$ denote a function class and let $\ell : [-1,1]^{\times 2} \to [0,1]$ denote a bounded loss function convex in the first argument and let $\mathcal{D}$ denote a dataset of size $n$. For $\eta > 0$, let $\bar{f}$ be as in Line 5 of Algorithm 2. Then for any $\beta > 0$, with probability at least $1 - \beta$,*

$$L(\bar{f}) - \inf_{f \in \mathcal{F}} L(f) \le \frac{12}{\sqrt{n}} \cdot \mathbb{E}\left[\sup_{f \in \mathcal{F}} \omega_n(f)\right] + 2 \cdot \sqrt{\frac{\log\left(\frac{1}{\beta}\right)}{n}} + 2\eta \cdot \left(\mathbb{E}\left[\sup_{f \in \mathcal{F}} \omega_m^{(j)}(f)\right] + \sqrt{\frac{\log\left(\frac{1}{\beta}\right)}{J}}\right).$$

(11)

*If instead we let $\bar{f}$ be as in Line 3 of Algorithm 3, then almost surely,*

$$L(\bar{f}) - \inf_{f \in \mathcal{F}} L(f) \le 12 \cdot \mathbb{E}\left[\sup_{f \in \mathcal{F}} \omega_n(f)\right] + 2 \cdot \sqrt{\frac{\log\left(\frac{1}{\beta}\right)}{n}} + \eta.$$

(12)

*Proof.* Applying Lemma 5 and noting that $\bar{f}, f_{\mathsf{ERM}} \in \mathcal{F}$ and applying Lemma 13 yields

$$L(\bar{f}) - \inf_{f \in \mathcal{F}} L(f) \le \frac{12}{\sqrt{n}} \cdot \mathbb{E}\left[\sup_{f \in \mathcal{F}} \omega_n(f)\right] + 2 \cdot \sqrt{\frac{2\log\left(\frac{1}{\beta}\right)}{n}} + \sup_{f,f' \in \mathcal{F}} R(f) - R(f').$$

The second statement follows immediately by noting that $0 \le \|f\|_m \le 1$ for all $f \in \mathcal{F}$ and letting $R(f) = \eta \cdot \|f\|_m$.

For the first statement, we note that by convexity of $\ell$, it holds that

$$L_{\mathcal{D}}(\bar{f}) \le \frac{1}{J} \cdot \sum_{j=1}^{J} L_{\mathcal{D}}(\bar{f}_j) \le L_{\mathcal{D}}(f_{\mathsf{ERM}}) + \frac{\eta}{J} \cdot \sum_{j=1}^{J} \sup_{f \in \mathcal{F}} \omega^{(j)}(f) - \inf_{f' \in \mathcal{F}} \omega^{(j)}(f').$$

To prove the second statement, we observe that by the Borell-Tsirelson-Ibragimov-Sudakov inequality (see, e.g., Wainwright [2019, Example 2.30]) and the fact that $\mathbb{E}\left[\omega_m^{(j)}(f)^2\right] \le 1$ for all $f \in \mathcal{F}$, that for all $j \in [J]$, with probability at least $1 - \beta$, it holds that

$$\sup_{f \in \mathcal{F}} \omega_m^{(j)}(f) \le \mathbb{E}\left[\sup_{f \in \mathcal{F}} \omega_m^{(j)}(f)\right] + \sqrt{2\log\left(\frac{1}{\beta}\right)}.$$

Applying symmetry and a Chernoff bound tells us that with probability at least $1 - \beta$ it holds that

$$\frac{1}{J} \cdot \sum_{j=1}^{J} \sup_{f \in \mathcal{F}} \omega_m^{(j)}(f) - \inf_{f \in \mathcal{F}} \omega_m^{(j)} \le 2 \cdot \mathbb{E}\left[\sup_{f \in \mathcal{F}} \omega_m^{(j)}(f)\right] + 2 \cdot \sqrt{\frac{\log\left(\frac{1}{\beta}\right)}{J}}.$$

The result follows. □

Before continuing, we prove Lemma 5 from Section 4:

*Proof of Lemma 5.* For an arbitrary regularizer $R : \mathcal{F} \to \mathbb{R}$, if we let $\bar{f} \in \mathrm{argmin}_{f \in \mathcal{F}} L_{\mathcal{D}}(f) + R(f)$, then by definition $L_{\mathcal{D}}(\bar{f}) + R(\bar{f}) \le L_{\mathcal{D}}(f_{\mathsf{ERM}}) + R(f_{\mathsf{ERM}})$ and so

$$L_{\mathcal{D}}(\bar{f}) \le L_{\mathcal{D}}(f_{\mathsf{ERM}}) + \sup_{f,f' \in \mathcal{F}} R(f) - R(f'),$$

where $f_{\mathsf{ERM}} \in \mathrm{argmin}_{f \in \mathcal{F}} L_{\mathcal{D}}(f)$ is the ERM. □

Combining these three lemmas yields the following PAC learning guarantee for Algorithm 2.

**Lemma 15** (PAC Learning Guarantee for Algorithm 2). *Suppose that $\mathcal{F} : \mathcal{X} \to [-1, 1]$ is a function class and $\ell : [-1, 1]^{\times 2} \to [0, 1]$ is a bounded loss function $\lambda$-Lipschitz and convex in the first argument. Let $\mu \in \Delta(\mathcal{X})$ and suppose that $\nu$ is $\sigma$-smooth with respect to $\mu$. For any $\beta > 0$ it holds with probability at least $1 - \beta$ that*

$$
L(\widehat{f}) - \inf_{f \in \mathcal{F}} L(f) \leq \frac{12}{\sqrt{n}} \cdot \mathbb{E}\left[ \sup_{f \in \mathcal{F}} \omega_n(f) \right] + 2 \cdot \sqrt{\frac{\log\left(\frac{1}{\beta}\right)}{n}}
$$

$$
+ 2\eta \cdot \left( \mathbb{E}\left[ \sup_{f \in \mathcal{F}} \omega_m^{(j)}(f) \right] + \sqrt{\frac{\log\left(\frac{1}{\beta}\right)}{J}} \right) + \frac{4\lambda\gamma}{\sigma} \cdot \sqrt{\log\left(\frac{1}{\beta}\right)}
$$

$$
+ \frac{C\lambda}{\sigma} \cdot \left( \frac{\log^3(m)}{\sqrt{m}} \cdot \overline{\mathcal{G}}_m(\mathcal{F}) + \sqrt{\frac{\log\log(m) + \log\left(\frac{1}{\beta}\right)}{m}} \right).
$$

*for $\widehat{f}$ returned by Algorithm 2 with $\mathcal{Q} = \mathcal{N}(0, 1)$ and $\mathcal{D}$ a dataset of size $n$.*

*Proof.* We compute:

$$
L(\widehat{f}) = L(\bar{f}) + L(\widehat{f}) - L(\bar{f}) \leq L(\bar{f}) + \lambda \cdot \left\| \bar{f} - \widehat{f} \right\|_{\nu_X} \leq L(\bar{f}) + \frac{\lambda}{\sigma} \cdot \left\| \bar{f} - \widehat{f} \right\|_{\mu},
$$

where the first inequality uses Jensen's and the second uses the fact that $\nu$ is $\sigma$-smooth. We now observe that by Lemma 8, with probability at least $1 - \beta$,

$$
\left\| \bar{f} - \widehat{f} \right\|_{\mu} \leq 2\gamma \cdot \sqrt{\log\left(\frac{1}{\beta}\right)}.
$$

Combining this with Lemma 14 yields

$$
L(\widehat{f}) - \inf_{f \in \mathcal{F}} L(f) \leq \frac{12}{\sqrt{n}} \cdot \mathbb{E}\left[ \sup_{f \in \mathcal{F}} \omega_n(f) \right] + 2 \cdot \sqrt{\frac{\log\left(\frac{1}{\beta}\right)}{n}} + 2\eta \cdot \left( \mathbb{E}\left[ \sup_{f \in \mathcal{F}} \omega_m^{(j)}(f) \right] + \sqrt{\frac{\log\left(\frac{1}{\beta}\right)}{J}} \right)
$$

$$
+ \frac{4\lambda\gamma}{\sigma} \cdot \sqrt{\log\left(\frac{1}{\beta}\right)} + \frac{\lambda}{\sigma} \left( \left\| \widehat{f} - \bar{f} \right\|_{\mu} - \left\| \widehat{f} - \bar{f} \right\|_{m} \right).
$$

Applying Lemma 12 concludes the result. $\qquad\square$

Similarly, we have a result for Algorithm 3; note that while convexity of $\ell$ is *required* to demonstrate that Algorithm 2 is a PAC learner, although is irrelevant to the privacy guarantee in Lemma 10, the situation for Algorithm 3 is reversed in that convexity *is not required* to demonstrate that Algorithm 3 is a PAC learner while it is necessary for the privacy guarantee in Lemma 11.

**Lemma 16** (PAC Guarantees for Algorithm 3). *Suppose that $\mathcal{F} : \mathcal{X} \to [-1, 1]$ is a convex function class and $\ell : [-1, 1]^{\times 2} \to [0, 1]$ is a bounded loss function $\lambda$-Lipschitz in the first argument. Let $\mu \in \Delta(\mathcal{X})$ and suppose that $\nu$ is $\sigma$-smooth with respect to $\mu$. For any $\beta > 0$ it holds with probability at least $1 - \beta$ that*

$$
L(\widehat{f}) - \inf_{f \in \mathcal{F}} L(f) \leq \frac{12}{\sqrt{n}} \cdot \mathbb{E}\left[ \sup_{f \in \mathcal{F}} \omega_n(f) \right] + 2 \cdot \sqrt{\frac{\log\left(\frac{1}{\beta}\right)}{n}} + \eta + \frac{4\lambda\gamma}{\sigma} \cdot \sqrt{\log\left(\frac{1}{\beta}\right)}
$$

$$
+ \frac{C\lambda}{\sigma} \cdot \left( \frac{\log^3(m)}{\sqrt{m}} \cdot \overline{\mathcal{G}}_m(\mathcal{F}) + \sqrt{\frac{\log\log(m) + \log\left(\frac{1}{\beta}\right)}{m}} \right).
$$

*for $\widehat{f}$ returned by Algorithm 3 with $\mathcal{Q} = \mathcal{N}(0,1)$ and $\mathcal{D}$ a dataset of size $n$. Similarly, if we replace $\mathcal{Q} = Lap(1)$, then with probability at least $1 - \beta$,*

$$L(\widehat{f}) - \inf_{f \in \mathcal{F}} L(f) \leq \frac{12}{\sqrt{n}} \cdot \mathbb{E}\left[ \sup_{f \in \mathcal{F}} \omega_n(f) \right] + 2 \cdot \sqrt{\frac{\log\left(\frac{1}{\beta}\right)}{n}} + \eta + \frac{4\lambda\gamma}{\sigma} \cdot \log\left(\frac{1}{\beta}\right) \cdot \sqrt{m}$$

$$+ \frac{C\lambda}{\sigma} \cdot \left( \frac{\log^3(m)}{\sqrt{m}} \cdot \overline{\mathcal{G}}_m(\mathcal{F}) + \sqrt{\frac{\log\log(m) + \log\left(\frac{1}{\beta}\right)}{m}} \right).$$

*Proof.* The proof of the first statement is identical to that of Lemma 15 with the exception of replacing (11) by (12) in the invocation of Lemma 14. The second statement is also identical but now replacing (9) by (10) when applying Lemma 8. $\qquad\square$

With these results in hand, along with those from Appendix C.4, all that remains to conclude the proofs of the main theorems is to tune the hyperparameters and control the complexity terms. We do this in the next section.

## C.6   Concluding the Proofs of Theorems 2 and 3

In this section, we combine the results from Appendices C.4 and C.5 to prove the main theorems. The main theorems in the text, Theorems 2 and 3, follow immediately from the following two results. We begin by stating a more detailed version of Theorem 2:

**Theorem 6.** *Suppose that $\mathcal{F} : \mathcal{X} \to [-1,1]$ is a function class such that $\overline{\mathcal{G}}_m(\mathcal{F}) = O\left(\sqrt{d}\right)$ for some $d \in \mathbb{N}$ and $\ell : [-1,1]^{\times 2} \to [0,1]$ is convex and $\lambda$-Lipschitz in the first argument. If we let $\mathcal{Q} = \mathcal{N}(0,1)$ in Algorithm 1 and set*

$$\gamma = \Theta\left( \frac{\sigma\alpha}{\lambda \cdot \sqrt{\log\left(\frac{1}{\beta}\right)}} \right), \qquad \eta = \Theta(\frac{\alpha}{\sqrt{d}}), \qquad m = \widetilde{\Theta}\left( \frac{d \vee \log\left(\frac{1}{\beta}\right)}{\sigma^2\alpha^2} \cdot \lambda^2 \right)$$

$$J = \widetilde{\Omega}\left( \frac{d\log\left(\frac{1}{\beta}\right)}{\alpha^2} \vee \frac{\lambda^8 d^2 \log^4\left(\frac{1}{\beta}\right) \log\left(\frac{1}{\delta}\right)}{\sigma^8\alpha^8\varepsilon^2} \vee \frac{d^2\lambda^6 \log^3\left(\frac{1}{\beta}\right) \log^2\left(\frac{1}{\delta}\right)}{\sigma^6\alpha^6\varepsilon^2} \right)$$

*and*

$$n = \widetilde{\Omega}\left( \frac{\lambda^{12} d^5 \log^6\left(\frac{1}{\beta}\right)}{\sigma^{12}\varepsilon^3\alpha^{14}} \vee \frac{d^5\lambda^9 \log^{9/2}\left(\frac{1}{\beta}\right) \log^{3/2}\left(\frac{1}{\delta}\right)}{\sigma^9\varepsilon^3\alpha^{10}} \right),$$

*then Algorithm 2 is $(\varepsilon, \delta)$-differentially private and an $(\alpha, \beta)$-PAC learner with respect to any $\nu$ that is $\sigma$-smooth with respect to $\mu$.*

*Proof.* This follows by combining Lemma 10 with Lemma 15 and plugging in the parameter choices. $\qquad\square$

We also have a result for Algorithm 3:

**Theorem 7.** *Suppose that $\mathcal{F} : \mathcal{X} \to [-1,1]$ is a convex function class such that $\overline{\mathcal{G}}_m(\mathcal{F}) = O\left(\sqrt{d}\right)$ for some $d \in \mathbb{N}$ and $\ell : [-1,1]^{\times 2} \to [0,1]$ is convex and $\lambda$-Lipschitz in the first argument. If we let $\mathcal{Q} = \mathcal{N}(0,1)$ in Algorithm 1 and set*

$$\gamma = \Theta\left( \frac{\sigma\alpha}{\lambda \cdot \sqrt{\log\left(\frac{1}{\beta}\right)}} \right), \qquad \eta = \Theta(\alpha), \qquad m = \widetilde{\Theta}\left( \frac{d \vee \log\left(\frac{1}{\beta}\right)}{\sigma^2\alpha^2} \cdot \lambda^2 \right)$$

*and*

$$n = \widetilde{\Omega}\left(\frac{\lambda^5 d \log^2\left(\frac{1}{\beta}\right)}{\varepsilon\sigma^4\alpha^5} \vee \frac{\lambda^4 d \log^{3/2}\left(\frac{1}{\beta}\right)\log^{1/2}\left(\frac{1}{\delta}\right)}{\varepsilon\sigma^3\alpha^4}\right),$$

*then Algorithm 3 is $(\varepsilon,\delta)$-differentially private and an $(\alpha,\beta)$-PAC learner with respect to any $\nu$ that is $\sigma$-smooth with respect to $\mu$.*

*On the other hand, if we set $\mathcal{Q} = Lap(1)$ in Algorithm 1 and set*

$$\gamma = \widetilde{\Theta}\left(\frac{\alpha^2\sigma}{\lambda^2 \cdot \left(d\sqrt{\log\left(\frac{1}{\beta}\right)} \wedge \log\left(\frac{1}{\beta}\right)\right)}\right), \quad \eta = \Theta(\alpha), \quad m = \widetilde{\Theta}\left(\frac{d \vee \log\left(\frac{1}{\beta}\right)}{\sigma^2\alpha^2} \cdot \lambda^2\right)$$

*and*

$$n = \widetilde{\mathcal{L}}\left(\frac{\lambda^6}{\sigma^5\varepsilon\alpha^6} \cdot \left(d^2\sqrt{\log\left(\frac{1}{\beta}\right)} \vee \log^{5/2}\left(\frac{1}{\beta}\right)\right)\right)$$

*then Algorithm 3 is $\varepsilon$-purely differentially private and an $(\alpha,\beta)$-PAC learner with respect to any $\sigma$-smooth $\nu$.*

*Proof.* This follows immediately by combining Lemma 11 with Lemma 16 and plugging in the parameter choices, with the pure differential privacy guarantees coming from the second halves of each lemma. □

As a final remark, we note that Lemmas 15 and 16 are both phrased entirely in terms of $\overline{\mathcal{G}}_m(\mathcal{F})$ and thus apply to function classes $\mathcal{F}$ such that $\overline{\mathcal{G}}_m(\mathcal{F}) = \omega(1)$. Such *non-donsker* [Wainwright, 2019, Van Handel, 2014] classes can still be learned in the PAC framework, albeit with slower rates. Indeed, it is immediate from the above results that as long as $\overline{\mathcal{G}}_m(\mathcal{F}) = o(\sqrt{m})$, then appropriately tuning the hyperparamters results Algorithms 2 and 3 being differentially private PAC learners with respect to $\nu$. It is well-known that (with our scaling) $\overline{\mathcal{G}}_m(\mathcal{F}) = o(\sqrt{m})$ is a necessary condition for PAC learnability even absent a privacy condition [Wainwright, 2019, Van Handel, 2014] and thus our results qualitatively demonstrate that private learnability with public data is possible whenever non-private learning is possible.

## C.7  Proof of Lemma 12

Replacing $\mathcal{F}$ by $\mathcal{F} - \mathcal{F} = \{f - f' | f, f' \in \mathcal{F}\}$ and noting that the uniform bound only increases by a factor of 2 and the Rademacher complexity increases at most by a factor of 2, we observe that it suffices to prove the result for $f' = 0$. We thus instead prove the notationally simpler claim that with probability at least $1 - \beta$, for all $f \in \mathcal{F}$,

$$\|f\|_\mu \leq 2 \cdot \|f\|_m + C\left(\frac{\log^3(m)}{\sqrt{m}} \cdot \overline{\mathcal{G}}_m(\mathcal{F}) + \frac{\log\log(m) + \log\left(\frac{1}{\beta}\right)}{m}\right)$$

We first note that by Bousquet [2002, Theorem 6.1], with probability at least $1 - \beta$, it holds that for all $f \in \mathcal{F}$,

$$\|f\|_\mu^2 \leq 2 \cdot \|f\|_m^2 + 200\left(\overline{r}^2 + \frac{\log\left(\frac{1}{\beta}\right) + \log\log(m)}{m}\right) \tag{13}$$

for some universal constant $C$, with

$$\overline{r} \leq \inf\left\{r > 0 \,\middle|\, \mathbb{E}_\xi\left[\sup_{\substack{f \in \mathcal{F} \\ \|f\|_m^2 \leq r^2}} \frac{1}{m} \cdot \sum_{i=1}^m \xi_i f(Z_i)^2\right] \leq \frac{r^2}{2}\right\}. \tag{14}$$

Taking square roots on both sides of (13) shows that it suffices to upper bound $\overline{r}$. For the remainder of the proof, we do this.

In order to proceed, we recall the following standard definition of covering numbers.

**Definition 8.** Let $\mathcal{F}$ be a function class and $\|\cdot\|_{m,\infty}$ the the $L^\infty$ norm on the empirical measure on $Z_1, \ldots, Z_m \in \mathcal{X}$, i.e., $\|f\|_{m,\infty} = \max_{i \in [m]} |f(Z_i)|$. We say that $f_1, \ldots, f_N$ is an $\varepsilon$-cover with respect to $\|\cdot\|_{m,\infty}$ if for all $f \in \mathcal{F}$ there is some $f_j$ such that $\|f - f_j\|_{m,\infty} \leq \varepsilon$. We then let $\mathcal{N}_{m,\infty}(\mathcal{F}, \varepsilon)$ denote the size of the smallest $\varepsilon$-cover of $\mathcal{F}$ with respect to $\|\cdot\|_{m,\infty}$.

The notion of a cover is standard throughout learning theory and can be used to control the Rademacher and Gaussian complexities [Dudley, 1969, Van Handel, 2014, Wainwright, 2019]. We will use it to control $\overline{r}$.

Proceeding with the proof, let $\mathcal{N}_{m,\infty}(\mathcal{F}, u)$ denote the covering number of the function class $\mathcal{F}$ with respect to $\|\cdot\|_{m,\infty}$ at scale $u > 0$. We then claim that $\overline{r}$ can be upper bounded by any $r$ satisfying

$$\frac{50}{\sqrt{m}} \cdot \int_{r/16}^1 \sqrt{\log \mathcal{N}_{m,\infty}(\mathcal{F}, u)} du \leq r. \tag{15}$$

We also claim that for any $r > \overline{\mathcal{G}}_m(\mathcal{F})$, the following holds:

$$\int_{r/16}^1 \sqrt{\log \mathcal{N}_{m,\infty}(\mathcal{F}, u)} du \leq C \sqrt{\log(m)} \cdot \left( \int_r^1 \frac{\sqrt{\log\left(\frac{cm}{u}\right)}}{u} du \right) \cdot \overline{\mathcal{G}}_m(\mathcal{F}) \tag{16}$$

for some universal constant $C$. We now suppose that (15) and (16) hold and set $r = Cm^{-1/2} \cdot \log^3(m) \cdot \overline{\mathcal{G}}_m(\mathcal{F})$. Then it is immediate that $r$ is a member of the set in (14).

We now prove the two claims.

**Proof that a solution to (15) is an upper bound on $\overline{r}$.** By a standard Dudley Chaining argument [Van Handel, 2014, Wainwright, 2019], it holds that

$$\mathbb{E}_\xi \left[ \sup_{\substack{f \in \mathcal{F} \\ \|f\|_m^2 \leq r^2}} \frac{1}{m} \cdot \sum_{i=1}^m \xi_i f(Z_i)^2 \right] \leq \inf_{u > 0} \left\{ 4u + \frac{12}{\sqrt{m}} \cdot \int_u^r \sqrt{\log \mathcal{N}_{m,\infty}(\mathcal{F}^2 \cap \left\{ \|f\|_m^2 \leq r^2 \right\}, u)} du \right\}. \tag{17}$$

Letting $f_1, \ldots, f_M \in \mathcal{F} \cap \left\{ \|f\|_M^2 \leq r^2 \right\}$ be a *proper* $u$-cover of $\mathcal{F} \cap \left\{ \|f\|_m^2 \leq r^2 \right\}$ with respect to $\|\cdot\|_{m,\infty}$ at scale $s \leq r$ and $\pi : \mathcal{F} \to \{f_i\}$ be projection to the cover, we have

$$\left\| f^2 - \pi(f)^2 \right\|_m^2 \leq s^2 \cdot \|f + \pi(f)\|_m^2 \leq 4s^2 r^2$$

by factoring $f^2 - \pi(f)^2 = (f - \pi(f))(f + \pi(f))$. In particular,

$$\mathcal{N}_{m,2}(\mathcal{F}^2 \cap \left\{ \|f\|_m^2 \leq r^2 \right\}, 2ur) \leq \mathcal{N}_{m,\infty}(\mathcal{F} \cap \left\{ \mathcal{F}^2 \cap \left\{ \|f\|_m^2 \leq r^2 \right\} \right\}, u) \leq \mathcal{N}_{m,\infty}(\mathcal{F}, u),$$

where we used the fact that a proper covering at scale $\varepsilon$ has size bounded by a covering at scale $\varepsilon/2$ by the triangle inequaltiy. Substituting into (17) and rescaling yields the claim.

**Proof that (16) holds.** This proof goes through fat shattering numbers, a complexity measure taking a function class and a scale $u > 0$ and returns $\mathsf{fat}(\mathcal{F}, u) \in \mathbb{N}$ [Bartlett et al., 1994]. We do not need the full definition of fat shattering numbers and defer to [Bartlett et al., 1994, Srebro et al., 2010, Rudelson and Vershynin, 2006] for details. We only need the following two properties. First, for any $u > 0$, it holds by Rudelson and Vershynin [2006] that

$$\log \mathcal{N}_{m,\infty}(\mathcal{F}, u) \leq C \cdot \mathsf{fat}(\mathcal{F}, cu) \cdot \log(m) \cdot \log\left( \frac{m}{\mathsf{fat}(\mathcal{F}, cu)u} \right). \tag{18}$$

Second, by Srebro et al. [2010, Lemma A.2], for all $r > m^{-1/2} \cdot \overline{\mathcal{G}}_m(\mathcal{F})$,

$$r^2 \cdot \mathsf{fat}(\mathcal{F}, r) \leq 4 \cdot \overline{\mathcal{G}}_m(\mathcal{F})^2. \tag{19}$$

Plugging (18) into the left hand side of (16) and then applying (19) concludes the proof of the claim.

# D  Classification and Analysis of RRSPM

In this section, we provide full proofs for the guarantees of Algorithm 4. In Section D.1, we describe the concept of universal identification sets and the result of Neel et al. [2019] which plays a crucial role in the privacy analysis of our algorithm. In Section D.2, we formally prove Lemma 6 which is the technical lemma used for the proof of differential privacy based on Neel et al. [2019]. We then continue in Section D.3 by applying standard learning theoretic techniques to demonstrate that our algorithm is an accurate classifier.

## D.1  Universal Identification Set based Algorithm

In this section, we formally define universal identification sets and informally describe the algorithm used in Neel et al. [2019]. Intuitively, a universal identification set captures the combinatorial property of a function class that all distinct functions in the function class disagree on at least one point from the data universe. For many natural classes, the size of the universal identification set is proportional to the VC dimension of the function class.

We now describe the algorithm from Neel et al. [2019] and explain the usefulness of universal identification sets. First, we formally define the notion of universal identification set.

**Definition 9.** (Universal Identification Set) A set $\mathcal{U} \subseteq \mathcal{X}$ is a universal identification set for a hypothesis class $\mathcal{F}$ if for all pairs of functions $f, f'$ in the hypothesis class $\mathcal{F}$, there is a $x \in \mathcal{U}$ such that:

$$f(x) \neq f'(x).$$

Additionally, if $|\mathcal{U}| = m$, we say that $\mathcal{F}$ has a universal identification set of size $m$.

Assuming the existence of a universal identification set for the function class $\mathcal{F}$ of size $m$ denoted by $\mathcal{U} = \{U_1, \ldots, U_m\}$, Neel et al. [2019] showed that the following algorithm(called RSPM) is an $\varepsilon$- pure differentially private and $(\alpha, \beta)$-accurate algorithm.

---

**Algorithm 5:** RSPM

---

1: **Input** ERM oracle ERM, dataset $\mathcal{D} = \{(X_i, Y_i) \mid 1 \leq i \leq n\}$, hypothesis class $\mathcal{F}$, universal identification set $\mathcal{U} = \{U_1, \ldots, U_m\}$, loss function $\ell : \mathcal{Y} \times \mathcal{Y} \to \{0, 1\}$.
2: **Draw** weights $\boldsymbol{\xi} = \{\xi_1, \ldots, \xi_m\}$ such that $\xi_i \sim \text{Lap}(2m/\varepsilon)$.
3: **Draw** labels $\tilde{Y} = \{\tilde{Y}_1, \ldots, \tilde{Y}_m\}$ such that $\tilde{Y}_i \sim \text{Uni}(\{0, 1\})$.
4: **Define** $\mathcal{L}_{\boldsymbol{\xi}, \mathcal{D}, \mathcal{U}} : \mathcal{F} \to \mathbb{R}$ such that

$$\mathcal{L}_{\boldsymbol{\xi}, \mathcal{D}, \mathcal{U}}(f) = \sum_{i=1}^{n} \ell(f(X_i), Y_i) + \sum_{i=1}^{m} \xi_i \cdot \ell(f(U_i), \tilde{Y}_i).$$

5: **Get** $\hat{f} = \text{ERM}(\mathcal{F}, \mathcal{L}_{\boldsymbol{\xi}, \mathcal{D}, \mathcal{U}})$.

---

RSPM roughly simulates "Report-Noisy-Min"(Dwork et al. [2014]) attempting to output a function that minimizes a perturbed estimate, where the perturbation is sampled from a Laplace distribution. A straight forward implementation of "Report-Noisy-Min" to minimize over all perturbed estimates of functions, it'd have to check for all functions in $\mathcal{F}$ and thus the computational complexity would depend on the size of $\mathcal{F}$. RSPM avoids this problem by implicitly perturbing the function evaluations via an augmented dataset. The proof of privacy thus exploits the structure of the universal identification set.

Although many natural function classes have bounded universal identification sets, their existence is not as general as having bounded VC dimension. In our work, we only assume finite VC dimension of the function class.

## D.2  Privacy Analysis

In this section, we prove that Algorithm 4 is differentially private. A notion that we will need is that of a projection of a hypothesis class onto a set of points from the domain.

**Definition 10.** (Projection) Given a hypothesis class $\mathcal{F} \subseteq \mathcal{Y}^{|\mathcal{X}|}$ and a subset $Z = \{z_1, \dots z_m\}$ of the feature space $\mathcal{X}$, we define the projection of $\mathcal{F}$ onto $Z$ to be $\mathcal{F}|_Z = \{(f(z_1), \dots, f(z_m)) : f \in \mathcal{F}\}$.

In the following sections, we will interchangeably think of the projection of a hypothesis class onto a set of points $Z$ as a set of functions on $Z$ or as a set of vectors in $\mathcal{Y}^{|Z|}$.

By construction, $\mathcal{F}|_Z$ has the property that any two distinct functions $f \neq f' \in \mathcal{F}|_Z$ must disagree on at least one point $z \in Z$. We encapsulate this as the following lemma.

**Lemma 17.** *Let $\mathcal{F}$ be a hypothesis class and let $Z = \{z_1, \dots, z_m\}$ be a set of points in the instance space $\mathcal{X}$. Then, for all $f \neq f' \in \mathcal{F}|_Z$, there exists $z \in Z$ such that $f(z) \neq f'(z)$.*

**Remark 2.** This property is analogous to the notion of a universal identification set considered in Neel et al. [2019]. In particular, the above lemma can be seen as the statement that the set $Z$ is a universal identification set for the class $\mathcal{F}|_Z$.

We first provide an informal sketch of the proof of privacy. In the later sections, we formalize these ideas. Let $\hat{f} \in \mathcal{F}$ be any arbitrary function and let $\mathcal{D}, \mathcal{D}'$ be any pair of neighbouring datasets. We show that $\mathbb{P}(\text{RRSPM}(\mathcal{D}) = \hat{f}) \leq e^\varepsilon \mathbb{P}(\text{RRSPM}(\mathcal{D}') = \hat{f})$. By the definition of the projection, there is some $\tilde{f} \in \mathcal{F}|_{\tilde{\mathcal{D}}_x}$ that is consistent with the labelling of the selected function $\hat{f}$. $\mathbb{P}(\text{ERM}(\mathcal{F}, \mathcal{L}_{\boldsymbol{\xi}, \mathcal{D}, \tilde{\mathcal{D}}}) = \tilde{f}) \leq e^\varepsilon \mathbb{P}(\text{ERM}(\mathcal{F}, \mathcal{L}_{\boldsymbol{\xi}, \mathcal{D}', \tilde{\mathcal{D}}}) = \tilde{f})$ since privacy for $\hat{f}$ follows from the post-processing property of differential privacy.

We now provide with a proof sketch for the main technical lemma showing that the privacy is preserved over the projected function class. Optimizing a loss function perturbed by Laplace-weighted examples implicitly tries to implement "Report-Noisy-Min" algorithm outputting a function that minimizes a perturbed estimate. For any neighbouring datasets $\mathcal{D}$ and $\mathcal{D}'$, the evaluation of any function $\tilde{f}$ can differ by at most 1. We show that the set of public points $\tilde{\mathcal{D}}$ is a universal identification set for the set of functions projected onto $\tilde{\mathcal{D}}$ and leverage this to prove that whenever the shift in the noise vectors is bounded by 2 in every coordinate, then $\tilde{f}$ is the minimizing function when switching from $\mathcal{D}$ to $\mathcal{D}'$. This intuition is made precise in Lemma 18.

**Lemma 18.** *Let $\mathcal{D}, \mathcal{D}'$ be two neighbouring data sets, and let $\tilde{\mathcal{D}} = (\tilde{\mathcal{D}}_x, \tilde{\mathcal{D}}_y) \in (\mathcal{X} \times \{0,1\})^m$ where $\tilde{\mathcal{D}}_x = \{Z_1, \dots, Z_m\}$ and $\tilde{\mathcal{D}}_y = \{\tilde{Y}_1, \dots, \tilde{Y}_m\}$. Define $\mathcal{E}(f_{\tilde{\mathcal{D}}}, \mathcal{D}, \tilde{\mathcal{D}}) = \left\{ \boldsymbol{\xi} : \text{ERM}(\mathcal{F}|_{\tilde{\mathcal{D}}_x}, \mathcal{L}_{\boldsymbol{\xi}, \mathcal{D}, \tilde{\mathcal{D}}}) = f_{\tilde{\mathcal{D}}} \right\}$, where $\mathcal{L}_{\boldsymbol{\xi}, \mathcal{D}, \tilde{\mathcal{D}}}$ is a functional as defined below:*

$$\mathcal{L}_{\boldsymbol{\xi}, \mathcal{D}, \tilde{\mathcal{D}}}(f) = \sum_{i=1}^{n} \ell(f(X_i), Y_i) + \sum_{i=1}^{m} \xi_i \cdot \ell(f(Z_i), \tilde{Y}_i).$$

*Let $\boldsymbol{\xi} = \{\xi_1, \dots, \xi_m\}$ such that $\xi_i \sim \text{Lap}(2m/\varepsilon)$. Given a fixed $f_{\tilde{\mathcal{D}}} \in \mathcal{F}|_{\tilde{\mathcal{D}}_x}$, define a mapping $\Psi_{f_{\tilde{\mathcal{D}}}}(\boldsymbol{\xi}) : \mathbb{R}^m \to \mathbb{R}^m$ on noise vectors as follows:*

1. *if $\ell(f_{\tilde{\mathcal{D}}}(Z_i), \tilde{Y}_i) = 1, \Psi_{f_{\tilde{\mathcal{D}}}}(\boldsymbol{\xi})_i = \xi_i - 2$*

2. *if $\ell(f_{\tilde{\mathcal{D}}}(Z_i), \tilde{Y}_i) = 0, \Psi_{f_{\tilde{\mathcal{D}}}}(\boldsymbol{\xi})_i = \xi_i + 2$*

*Equivalently, $\Psi_{f_{\tilde{\mathcal{D}}}}(\boldsymbol{\xi})_i = \xi_i + 2(1 - 2\ell(f_{\tilde{\mathcal{D}}}(Z_i), \tilde{Y}_i))$. Let $\boldsymbol{\xi} \in \mathcal{E}\left(f_{\tilde{\mathcal{D}}}, \mathcal{D}, \tilde{\mathcal{D}}\right)$ where $f_{\tilde{\mathcal{D}}} \in \text{ERM}(\mathcal{F}|_{\tilde{\mathcal{D}}_x}, \mathcal{L}_{\boldsymbol{\xi}, \mathcal{D}, \tilde{\mathcal{D}}})$. Then $\Psi_{f_{\tilde{\mathcal{D}}}}(\boldsymbol{\xi}) \in \mathcal{E}(f_{\tilde{\mathcal{D}}}, \mathcal{D}', \tilde{\mathcal{D}})$.*

*Proof.* Let $\Psi_{f_{\tilde{\mathcal{D}}}}(\boldsymbol{\xi}) = \boldsymbol{\xi}' = (\xi_1', \dots \xi_m')$. Our goal is to show that for every $f \in \mathcal{F}|_{\tilde{\mathcal{D}}_x}$ such that $f \neq f_{\tilde{\mathcal{D}}}$, we have $\mathcal{L}_{\boldsymbol{\xi}', \mathcal{D}', \tilde{\mathcal{D}}}(f) > \mathcal{L}_{\boldsymbol{\xi}', \mathcal{D}', \tilde{\mathcal{D}}}(f_{\tilde{\mathcal{D}}})$. First, recall that by our assumption for all $f \in \mathcal{F}|_{\tilde{\mathcal{D}}_x}$, we have

$$\mathcal{L}_{\boldsymbol{\xi}, \mathcal{D}, \tilde{\mathcal{D}}}(f) > \mathcal{L}_{\boldsymbol{\xi}, \mathcal{D}, \tilde{\mathcal{D}}}(f_{\tilde{\mathcal{D}}}). \tag{20}$$

We now argue that $\mathcal{L}_{\boldsymbol{\xi}',\mathcal{D}',\tilde{\mathcal{D}}}(f)-\mathcal{L}_{\boldsymbol{\xi}',\mathcal{D}',\tilde{\mathcal{D}}}(f_{\tilde{\mathcal{D}}})$ is strictly positive for all $f \in \mathcal{F}|_{\tilde{\mathcal{D}}_x}$ such that $f_{\tilde{\mathcal{D}}} \neq f$. To see this we calculate,

$$
\begin{aligned}
\mathcal{L}_{\boldsymbol{\xi}',\mathcal{D}',\tilde{\mathcal{D}}}(f) - \mathcal{L}_{\boldsymbol{\xi}',\mathcal{D}',\tilde{\mathcal{D}}}(f_{\tilde{\mathcal{D}}}) = &\sum_{(X,Y)\in\mathcal{D}'} \ell(f(X),Y) + \sum_{i=1}^{m} \xi_i' \cdot \ell(f(Z_i),\tilde{Y}_i) \\
&- \sum_{(X,Y)\in\mathcal{D}'} \ell(f_{\tilde{\mathcal{D}}}(X),Y) - \sum_{i=1}^{m} \xi_i' \cdot \ell(f_{\tilde{\mathcal{D}}}(Z_i),\tilde{Y}_i) \\
\geq\ & \mathcal{L}_{\boldsymbol{\xi},\mathcal{D},\tilde{\mathcal{D}}}(f) - 1 + \sum_{i=1}^{m} \xi_i' \cdot \ell(f(Z_i),\tilde{Y}_i) - \xi_i \cdot \ell(f(Z_i),\tilde{Y}_i) \\
&- \mathcal{L}_{\boldsymbol{\xi},\mathcal{D},\tilde{\mathcal{D}}}(f_{\tilde{\mathcal{D}}}) - 1 - \sum_{i=1}^{m} \xi_i' \cdot \ell(f_{\tilde{\mathcal{D}}}(Z_i),\tilde{Y}_i) + \xi_i \cdot \ell(f_{\tilde{\mathcal{D}}}(Z_i),\tilde{Y}_i) \\
>\ & -2 + \sum_{i=1}^{m} (\xi_i' - \xi_i)\left(\ell(f(Z_i),\tilde{Y}_i)) - \ell(f_{\tilde{\mathcal{D}}}(Z_i),\tilde{Y}_i)\right),
\end{aligned}
$$

where the second inequality follows from the fact that $\mathcal{D}$ and $\mathcal{D}'$ differ in only one entry and $\ell$ is $1-$ sensitive. The last equation follows from statement 20.

We know from Lemma 17 that there exists a $Z \in \tilde{\mathcal{D}}$ such that $f(Z) \neq f_{\tilde{\mathcal{D}}}(Z)$. Recall that $\xi_i' = \xi_i + 2(1 - 2\ell(f_{\tilde{\mathcal{D}}}(Z_i),\tilde{Y}_i))$. By construction, each term is non-negative. Therefore,

$$
\sum_{i=1}^{m} (\xi_i' - \xi_i)\left(\ell(f(Z_i),\tilde{Y}_i)) - \ell(f_{\tilde{\mathcal{D}}}(Z_i),\tilde{Y}_i)\right) > 2.
$$

To wrap up, we can bound

$$
\mathcal{L}_{\boldsymbol{\xi}',\mathcal{D}',\tilde{\mathcal{D}}}(f) - \mathcal{L}_{\boldsymbol{\xi}',\mathcal{D}',\tilde{\mathcal{D}}}(f_{\tilde{\mathcal{D}}}) > 0.
$$

This proves that $\Psi_{f_{\tilde{\mathcal{D}}}}(\boldsymbol{\xi}) \in \mathcal{E}(f_{\tilde{\mathcal{D}}}, \mathcal{D}', \tilde{\mathcal{D}})$.

$\square$

**Lemma 19** (Laplace shift). *Let* $\boldsymbol{\xi} = \{\xi_1,\ldots,\xi_m\}$ *such that* $\xi_i \sim \mathrm{Lap}(2m/\varepsilon)$. *Fix some noise realization* $\mathbf{r} \in \mathbb{R}^m$ *and fix a hypothesis in the projection set* $f \in \mathcal{F}|_{\tilde{\mathcal{D}}_x}$. *Then,*

$$
\mathbb{P}(\boldsymbol{\xi} = \mathbf{r}) \leq e^{\varepsilon} \mathbb{P}(\boldsymbol{\xi} = \Psi_f(\mathbf{r})),
$$

*where* $\Psi_f(\boldsymbol{\xi})_i = \xi_i + 2(1 - 2\ell(f(Z_i),\tilde{Y}_i))$ *as defined in Lemma 18.*

*Proof.* Let $i \in [m]$ be any index and let $\mathbf{r} \in \mathbb{R}^m$. Since $\xi_i \sim \mathrm{Lap}(2m/\varepsilon)$, we know

$$
\mathbb{P}(\xi_i = r_i) = \frac{\varepsilon}{4m} \exp\left(\frac{-|r_i|\varepsilon}{2m}\right)
$$

For any $t, t' \in \mathbb{R}$ such that $|t - t'| \leq 2$, we get

$$
\mathbb{P}(\xi_i = t') \leq \exp(\varepsilon/m)\mathbb{P}(\xi_i = t)
$$

Since for all $i \in [d]$, $|\Psi_f(r)_i - r_i| \leq 2$, we get

$$
\frac{\mathbb{P}(\boldsymbol{\xi} = \Psi_f(\mathbf{r}))}{\mathbb{P}(\boldsymbol{\xi} = \mathbf{r})} = \prod_{i=1}^{m} \frac{\mathbb{P}(\xi_i = \Psi_f(r)_i)}{\mathbb{P}(\xi_i = r_i)} \leq \prod_{i=1}^{m} \exp(\varepsilon/m) = \exp(\varepsilon).
$$

$\square$

Our proof of privacy also makes use of the following lemma, which says that minimizers are unique with probability 1 [Neel et al., 2019, Lemma 4].

**Lemma 20.** *Let $\tilde{\mathcal{D}} = (\tilde{\mathcal{D}}_x, \tilde{\mathcal{D}}_y) \in (\mathcal{X} \times \{0,1\})^m$. Consider $\mathcal{F}|_{\tilde{\mathcal{D}}_x}$, where $\mathcal{F}|_{\tilde{\mathcal{D}}_x}$ is the projection of $\mathcal{F}$ on $\tilde{\mathcal{D}}_x$. For every dataset $\mathcal{D}$, there is a subset $B \subseteq \mathbb{R}^m$ such that:*

- $\mathbb{P}(\boldsymbol{\xi} \in B) = 0$ *and*

- *On the restricted domain $\mathbb{R}^m \setminus B$, there is a unique minimizer $\tilde{f} \in \arg\min_{f \in \mathcal{F}|_{\tilde{\mathcal{D}}_x}} \mathcal{L}_{\boldsymbol{\xi}, \mathcal{D}, \tilde{\mathcal{D}}}(f)$.*

Using standard results about Laplace perturbations stated in Lemma 19 and Lemma 20 and using the perturbation coupling bound in Lemma 18, we get the our formal statement of privacy for optimizing over the projected function class as follows:

**Lemma 21.** *(Privacy over Projection) Let $\mathcal{D}, \mathcal{D}'$ be arbitrary datasets containing $n$ points each. Let $\tilde{\mathcal{D}} = (\tilde{\mathcal{D}}_x, \tilde{\mathcal{D}}_y) \in (\mathcal{X} \times \{0,1\})^m$. Then,*
$$\mathbb{P}(\mathsf{ERM}(\mathcal{F}|_{\tilde{\mathcal{D}}_x}, \mathcal{L}_{\boldsymbol{\xi}, \mathcal{D}, \tilde{\mathcal{D}}}) = f) \le e^\varepsilon \mathbb{P}(\mathsf{ERM}(\mathcal{F}|_{\tilde{\mathcal{D}}_x}, \mathcal{L}_{\boldsymbol{\xi}, \mathcal{D}', \tilde{\mathcal{D}}}) = f).$$

*Proof.* We calculate,

$$
\begin{aligned}
\mathbb{P}(\mathsf{ERM}(\mathcal{F}|_{\tilde{\mathcal{D}}_x}, \mathcal{L}_{\boldsymbol{\xi}, \mathcal{D}, \tilde{\mathcal{D}}}) = f) &= \mathbb{P}(\boldsymbol{\xi} \in \mathcal{E}(f, \mathcal{D}, \tilde{\mathcal{D}})) \\
&= \int_{\mathbb{R}^m} \mathbb{P}(\boldsymbol{\xi}) \mathbb{1}(\xi \in \mathcal{E}(f, \mathcal{D}, \tilde{\mathcal{D}})) d\boldsymbol{\xi} \\
&= \int_{\mathbb{R}^m \setminus B} \mathbb{P}(\boldsymbol{\xi}) \mathbb{1}(\xi \in \mathcal{E}(f, \mathcal{D}, \tilde{\mathcal{D}})) d\boldsymbol{\xi} && B \text{ has 0 measure by Lemma 20} \\
&\le \int_{\mathbb{R}^m \setminus B} \mathbb{P}(\boldsymbol{\xi}) \mathbb{1}(\Psi_f(\boldsymbol{\xi}) \in \mathcal{E}(f, \mathcal{D}', \tilde{\mathcal{D}})) d\boldsymbol{\xi} && \text{Lemma 18} \\
&\le \int_{\mathbb{R}^m \setminus B} e^\varepsilon \mathbb{P}(\Psi_f(\boldsymbol{\xi})) \mathbb{1}(\Psi_f(\boldsymbol{\xi}) \in \mathcal{E}(f, \mathcal{D}', \tilde{\mathcal{D}})) d\boldsymbol{\xi} && \text{Lemma 19} \\
&\le \int_{\mathbb{R}^m \setminus \Psi_f(B)} e^\varepsilon \mathbb{P}(\boldsymbol{\xi}) \mathbb{1}(\boldsymbol{\xi} \in \mathcal{E}(f, \mathcal{D}', \tilde{\mathcal{D}})) \frac{\partial \Psi_f}{\partial \boldsymbol{\xi}} d\boldsymbol{\xi} && \text{Change of variables } \boldsymbol{\xi} \to \Psi_f(\boldsymbol{\xi}) \\
&= \int_{\mathbb{R}^m} e^\varepsilon \mathbb{P}(\boldsymbol{\xi}) \mathbb{1}(\boldsymbol{\xi} \in \mathcal{E}(f, D', \tilde{\mathcal{D}})) d\boldsymbol{\xi} && \Psi_f(B) \text{ has 0 measure, } \left| \frac{\partial \Psi_f}{\partial \boldsymbol{\xi}} \right| = 1 \\
&= e^\varepsilon \mathbb{P}(\boldsymbol{\xi} \in \mathcal{E}(f, D', \tilde{\mathcal{D}})) \\
&= e^\varepsilon \mathbb{P}(\mathsf{ERM}(\mathcal{F}|_{\tilde{\mathcal{D}}_x}, \mathcal{L}_{\boldsymbol{\xi}, \mathcal{D}', \tilde{\mathcal{D}}}) = f).
\end{aligned}
$$

$\square$

**Theorem 8.** *Algorithm 4 is $\varepsilon$-pure differentially private.*

*Proof.* Let $\mathcal{D}$ and $\mathcal{D}'$ be any neighbouring datasets. Fix any function $\hat{f} \in \mathcal{F}$. We now show that
$$\mathbb{P}(\mathsf{RRSPM}(\mathcal{D}) = \hat{f}) \le e^\varepsilon \mathbb{P}(\mathsf{RRSPM}(\mathcal{D}') = \hat{f}),$$

where the probability is taken over the randomness of the algorithm. From the definition of the projection, we know that there exists a unique function in the projection say, $\tilde{f} \in \mathcal{F}|_{\tilde{\mathcal{D}}_x}$ such that $\hat{f}(Z_i) = \tilde{f}(Z_i)$ for all $i \in [m]$.

Using Lemma 21, we know that

$$\mathbb{P}(\mathsf{ERM}(\mathcal{F}|_{\tilde{\mathcal{D}}_x}, \mathcal{L}_{\boldsymbol{\xi}, \mathcal{D}, \tilde{\mathcal{D}}}) = \tilde{f}) \le e^\varepsilon \mathbb{P}(\mathsf{ERM}(\mathcal{F}|_{\tilde{\mathcal{D}}_x}, \mathcal{L}_{\boldsymbol{\xi}, \mathcal{D}', \tilde{\mathcal{D}}, \tilde{\mathcal{D}}_y}) = \tilde{f}).$$

As defined in the algorithm, let $\tilde{\mathcal{L}}(f) = \sum_{i=1}^m \ell(f(Z_i), \tilde{f}(Z_i))$. From the definition of $\tilde{\mathcal{L}}$, it follows that $\hat{f} \in \arg\min_{f \in \mathcal{F}} \tilde{\mathcal{L}}(f)$. Following the post-processing guarantee of differential privacy, it is easy to see that

$$\mathbb{P}(\mathsf{RRSPM}(\mathcal{D}) = \hat{f}) \leq e^{\varepsilon}\mathbb{P}(\mathsf{RRSPM}(\mathcal{D}') = \hat{f}).$$

$\square$

### D.3 Accuracy Analysis

In this section we analyze the accuracy of our algorithm. Let $f^*$ denote the function in the hypothesis class that minimizes the loss with respect to the distribution $\nu$ i.e. $f^* \in \arg\min\limits_{f \in \mathcal{F}} L_\nu(f)$. Let $\hat{f}$ denote the output hypothesis of our algorithm. We show that the loss of $\hat{f}$ is close to $f^*$ with respect to the data generating distribution $\nu$.

As in the algorithm description, let $\tilde{f}$ be the function that minimizes the perturbed loss. Our algorithm outputs $\hat{f}$ whose labelling is consistent with $\tilde{f}$ on the public dataset $\tilde{\mathcal{D}}_x$. Since $\tilde{\mathcal{D}}_x$ is sampled from the base distribution, it follows from VC theorem that $\hat{f}$ and $\tilde{f}$ are close under the base distribution $\mu$. We show in Lemma 22 that $\hat{f}$ and $\tilde{f}$ are close under $\nu$ by leveraging that $\nu_x$ is a $\sigma$-smooth distribution. Using standard results about Laplace perturbations, we show that $f'$ is close to $\tilde{f}$ in Lemma 23, where $f'$ is the empirical risk minimizer over $\mathcal{D}$. Using the VC theorem, it is easy to see that $f'$ is close to $f^*$ and consequently using the triangle inequality we finish the proof by showing that $f^*$ and $\hat{f}$ are close under $\nu$.

We now state the VC theorem below which we use in our analysis.

**Theorem 9.** *Let $\mathcal{D} = \{(X_1, Y_1), \ldots, (X_n, Y_n)\}$ where for all $i \in [n]$, $(X_i, Y_i) \in \mathcal{X} \times \{0, 1\}$ are sampled from a fixed distribution $\nu$. Let $L_D(f) = \frac{1}{n}|\{i : f(X_i) \neq Y_i\}|$ and let $L_\nu(f) = \mathbb{E}_{(X,Y) \sim \nu}[\mathbb{1}(f(X) \neq Y)]$. If the function class $\mathcal{F}$ has VC dimension $d$ then,*

$$\mathbb{P}\left(\sup_{f \in \mathcal{F}}|L_{\mathcal{D}}(f) - L_\nu(f)| \leq O\left(\sqrt{\frac{d + \log(1/\beta)}{m}}\right)\right) \geq 1 - \beta.$$

*In particular if $f^* \in \arg\min\limits_{f \in \mathcal{F}} L_{\mathcal{D}}(f)$ then,*

$$\mathbb{P}\left(|L_\nu(f^*) - \arg\min\limits_{f \in \mathcal{F}} L_\nu(f)| \leq O\left(\sqrt{\frac{d + \log(1/\beta)}{m}}\right)\right) \geq 1 - \beta.$$

**Lemma 22.** *Let $\nu_x$ be a $\sigma$-smooth distribution that is $\left\|\frac{d\nu_x}{d\mu}\right\| \leq \frac{1}{\sigma}$. Let $L_\nu$ be the loss function as defined in Defintion 2. Let $\hat{f}$ be the hypothesis returned by our algorithm and let $\tilde{f} \in \mathsf{ERM}(\mathcal{F}, \mathcal{L}_{\boldsymbol{\xi}, \mathcal{D}, \tilde{\mathcal{D}}})$. Then,*

$$L_\nu(\hat{f}) - L_\nu(\tilde{f}) \leq O\left(\sqrt{\frac{d + \log(1/\beta)}{m\sigma^2}}\right),$$

*with probability $1 - \beta$.*

*Proof.* Let $\alpha' = O\left(\sqrt{\frac{d + \log(1/\beta)}{m}}\right)$. First we show that $|\underset{\substack{X \sim \mu \\ Y = \tilde{f}(X)}}{L(\hat{f})} - \underset{\substack{X \sim \mu \\ Y = \tilde{f}(X)}}{L(\tilde{f})}| \leq 2\alpha'$ with probability at least $1 - \beta$. Consider the dataset $\hat{D} = \{(X_1, \tilde{f}(X_1)), \ldots, (X_m, \tilde{f}(X_m))\}$ we minimize over and output $\hat{f}$. Using Theorem 9 we know that,

$$\mathbb{P}\left(|\underset{\substack{X \sim \mu \\ Y = \tilde{f}(X)}}{L(\hat{f})} - \underset{\substack{X \sim \mu \\ Y = \tilde{f}(X)}}{L(\tilde{f})}| \leq 2\alpha' + |L_{\hat{D}}(\hat{f}) - L_{\hat{D}}(\tilde{f})|\right) \geq 1 - \beta.$$

Since $\underset{\substack{X \sim \mu \\ Y = \tilde{f}(X)}}{L(\tilde{f})} = 0$ and $|L_{\tilde{\mathcal{D}}}(\hat{f}) - L_{\tilde{\mathcal{D}}}(\tilde{f})| = 0$ and we get that with probability at least $1 - \beta$,

$$\mathbb{P}_{X \sim \mu}(\hat{f}(x) \neq \tilde{f}(x)) \leq 2\alpha'. \tag{21}$$

We now show that $\mathbb{P}_{X \sim \nu_x}(\hat{f}(x) \neq \tilde{f}(x)) \leq 2\alpha'/\sigma$ with probability at least $1 - \beta$.

$$
\begin{aligned}
\mathbb{P}_{X \sim \nu_x}(\hat{f}(x) \neq \tilde{f}(x)) &= \mathbb{E}_X \sim \nu_x[\mathbb{1}(\hat{f}(x) \neq \tilde{f}(x))] \\
&= \sum_{X \in \mathcal{X}} \mathbb{P}(\nu_x = X)\mathbb{1}(\hat{f}(x) \neq \tilde{f}(x)) \\
&\leq \frac{1}{\sigma} \sum_{X \in \mathcal{X}} \mathbb{P}(\mu = X)\mathbb{1}(\hat{f}(x) \neq \tilde{f}(x)) \qquad \text{Since } \left\|\frac{d\nu_x}{d\mu}\right\| \leq \frac{1}{\sigma} \\
&= \frac{1}{\sigma}\left(\mathbb{E}_X \sim \mu[\mathbb{1}(\hat{f}(x) \neq \tilde{f}(x))]\right) \\
&= \frac{1}{\sigma}\left(\mathbb{P}_{X \sim \mu}(\hat{f}(x) \neq \tilde{f}(x))\right) \\
&\leq \frac{2\alpha}{\sigma} \qquad \text{Using Eq 21}
\end{aligned}
$$

Having shown that with probability at least $1 - \beta$,

$$\mathbb{P}_{X \sim \nu_x}(\hat{f}(x) \neq \tilde{f}(x)) \leq 2\alpha'/\sigma, \tag{22}$$

we now prove that with probability at least $1 - \beta$,

$$\left| \underset{\substack{X \sim \nu_x \\ Y \sim \nu_y|X}}{L(\hat{f})} - \underset{\substack{X \sim \nu_x \\ Y \sim \nu_y|X}}{L(\tilde{f})} \right| \leq 2\alpha'/\sigma.$$

which is equivalent to the theorem statement. Using the triangle inequality we get

$$
\begin{aligned}
\left| \underset{\substack{X \sim \nu_x \\ Y \sim \nu_y|X}}{L(\hat{f})} - \underset{\substack{X \sim \nu_x \\ Y \sim \nu_y|X}}{L(\tilde{f})} \right| &\leq \mathbb{E}_{X \sim \nu_x}\left[\left|\mathbb{P}_{Y \sim \nu_y|X}(\hat{f}(X) \neq Y) - \mathbb{P}_{Y \sim \nu_y|X}(\tilde{f}(X) \neq Y)\right|\right] \\
&= \mathbb{E}_{X \sim \nu_x}[\mathbb{1}(\tilde{f}(X) \neq \hat{f}(X))] \\
&= 2\alpha'/\sigma
\end{aligned}
$$

where the second equation follows from the observation that for any value of $X$, if $\hat{f}(X) = \tilde{f}(X)$ then the difference of the probabilities equate to 0 and if $\hat{f}(X) \neq \tilde{f}(X)$ then the difference of the probabilities equate to 1 and the last equation follows from Equation 22.

Substituting the value of $\alpha'$ proves that with probability $1 - \beta$,

$$L_\nu(\hat{f}) - L_\nu(\tilde{f}) \leq O\left(\sqrt{\frac{d + \log(1/\beta)}{m\sigma^2}}\right).$$

$\square$

**Lemma 23.** *Let $\mathcal{F}$ be a function class with $VC$ dimension $d$. Let $\mathcal{D} = \{(X_1, Y_1), \ldots, (X_n, Y_n)\}$ and $\tilde{\mathcal{D}}_x = \{Z_1, \ldots, Z_m\}$ where $Z_i \sim \mu$. Let $L_\mathcal{D}$ be the loss function as defined in Definion 2. Let $f' \in \mathsf{ERM}(\mathcal{F}, \mathcal{L}_\mathcal{D})$ and let $\tilde{f} \in \mathsf{ERM}(\mathcal{F}, \mathcal{L}_{\boldsymbol{\xi}, \mathcal{D}, \tilde{\mathcal{D}}})$ then,*

$$L_\mathcal{D}(\tilde{f}) - L_\mathcal{D}(f') \leq \frac{4m^2 \log(m/\beta)}{\varepsilon n},$$

*with probability $1 - \beta$.*

*Proof.* Following the algorithm we know that for all $i \in [m], \xi_i \sim \mathrm{Lap}(2m/\varepsilon)$. Using Chernoff's bound and a union bound we get that with probability $1 - \beta$,

$$\forall i \in [m], |\xi_i| \leq \frac{2m \log(m/\beta)}{\varepsilon}.$$

Since $\tilde{f}(Z_i) \in \{0,1\}$ for all $Z_i \in \tilde{\mathcal{D}}$, with probability $1-\beta$, $\mathcal{L}_{\boldsymbol{\xi},\mathcal{D},\tilde{\mathcal{D}}}(\tilde{f}) \geq \mathcal{L}_{\mathcal{D}}(\tilde{f}) - m \cdot \frac{2m \log(m/\beta)}{\varepsilon}$.
Similarly, $\mathcal{L}_{\boldsymbol{\xi},\mathcal{D},\tilde{\mathcal{D}}}(f') \leq \mathcal{L}_{\mathcal{D}}(f') + m \cdot \frac{2m \log(m/\beta)}{\varepsilon}$. Dividing by $n$ and combining the bounds we get,

$$L_{\mathcal{D}}(\tilde{f}) - L_{\mathcal{D}}(f') \leq \frac{4m^2 \log(m/\beta)}{\varepsilon n},$$

with probability $1-\beta$. $\qquad\square$

**Theorem 10.** *Let $\hat{f}$ be as defined in Algorithm with $\mathcal{D}$ sampled from some distribution $\nu$ such that $\left\| \frac{d\nu_x}{d\mu} \right\| \leq \frac{1}{\sigma}$. Suppose the function class $\mathcal{F}$ has VC-dimension $d$ then setting*

$$m = O\left( \frac{d + \log(1/\beta)}{\alpha^2 \sigma^2} \right)$$

*yields an $\varepsilon$- pure differentially private $(\alpha, \beta)$-learner as long as*

$$n \geq \tilde{\Omega}\left( \frac{d^2 \log(1/\beta)}{\alpha^5 \sigma^4 \varepsilon} \right),$$

*where $\tilde{\Omega}$ hides* log *factors.*

*Proof.* Let $\nu$ be the distribution from which the given dataset $\mathcal{D}$ is sampled where $\left\| \frac{d\nu_x}{d\mu} \right\| \leq \frac{1}{\sigma}$. Let $\hat{f}$ be the output of our algorithm, $f' \in \arg\min_{f \in \mathcal{F}} \mathcal{L}_{\mathcal{D}}(f)$, $f^*$ be the target function from our function class $\mathcal{F}$ and $\tilde{f} \in \arg\min_{f \in \mathcal{F}} \mathcal{L}_{\boldsymbol{\xi},\mathcal{D},\tilde{\mathcal{D}}}(f)$. We wish to compare the guarantee of the function $\hat{f}$ with respect to the function $f^*$ with respect to the distribution $\nu$ i.e. $|L_\nu(\hat{h}) - L_\nu(f^*)|$.

In our analysis below, we break the loss function several times using triangle inequality and bound each term using previously stated results via a union bound.

$|L_\nu(\hat{f}) - L_\nu(f^*)|$
$\leq |L_\nu(\hat{f}) - L_\nu(\tilde{f})| + |L_\nu(\tilde{f}) - L_{\mathcal{D}}(\tilde{f})| + |L_{\mathcal{D}}(\tilde{f}) - L_{\mathcal{D}}(f')| + |L_{\mathcal{D}}(f') - L_\nu(f^*)|$
$\leq \underbrace{O\left( \sqrt{\frac{d + \log(1/\beta)}{\sigma^2 m}} \right)}_{\text{Lemma 22}} + \underbrace{O\left( \sqrt{\frac{d + \log(1/\beta)}{n}} \right)}_{\text{Theorem 9}} + \underbrace{O\left( \frac{m^2 \log(m/\beta)}{\varepsilon n} \right)}_{\text{Lemma 23}} + \underbrace{O\left( \sqrt{\frac{d + \log(1/\beta)}{n}} \right)}_{\text{Theorem 9}}$
$= O\left( \sqrt{\frac{d + \log(1/\beta)}{\sigma^2 m}} + \sqrt{\frac{d + \log(1/\beta)}{n}} + \frac{m^2 \log(m/\beta)}{\varepsilon n} \right).$

Setting

$$m = O\left( \frac{d + \log(1/\beta)}{\alpha^2 \sigma^2} \right), \; n = O\left( \frac{d^2 \log d \cdot \log(1/\beta)}{\alpha^5 \sigma^4 \varepsilon} \right)$$

and combining the guarantee from Theorem 8 we get that RRSPM (Algorithm 4) is a $\varepsilon$- pure differentially private $(\alpha, \beta)$-learner. $\qquad\square$

