# OpenReview forum: "Oracle-Efficient Differentially Private Learning with Public Data"
_NeurIPS.cc/2024/Conference — NeurIPS 2024 poster_

### Official Review · Reviewer_b8gT · 2024-07-04

**Soundness:** 2
**Presentation:** 2
**Contribution:** 3
**Rating:** 5
**Confidence:** 2

**Summary:**

This paper explores methods for private learning by leveraging public unlabeled data. Previous work in this area utilized public data to construct an $\alpha$-covering and then employed an exponential mechanism to output a hypothesis privately. However, the primary drawback of this approach was its exponential running time in both building the covering and sampling from the exponential mechanism. In contrast, this work assumes the availability of an Empirical Risk Minimization (ERM) oracle, which can find the minimizer of a function class given a loss function. The authors propose new private and oracle-efficient algorithms with polynomial complexity in terms of Gaussian complexity, objective error $\alpha$, and other parameters.

**Strengths:**

The inefficiency of previous methods highlights the importance of this problem. This paper makes significant progress by designing private algorithms that achieve oracle efficiency, addressing a crucial gap in the literature.

**Weaknesses:**

I find it hard to understand the intuitions behind the algorithm design and, hence, hard to appreciate the results, possibly because I am not an expert in learning theory. Several areas remain unclear to me:
(1). If we already have the strong convexity and assume the L is differentiable, can we compute the gradient, run the gradient descent, and reduce the problem to the convex optimization?
(2) What is the main motivation for considering Gaussian complexity rather than the VC dimension? Can the results from previous studies be extended to Gaussian complexity?
(3) There is a lack of discussion on sample complexity. For instance, Theorem 2 requires $n\ge \Omega(1/\alpha^{14})$. It is helpful to discuss the origin of this term.
(4) What is the state-of-the-art (SOTA) performance without public data? Does the proposed approach with public data outperform current methods in terms of sample complexity?

**Questions:**

Please refer to the Weaknesses above.

**Limitations:**

Yes.

---

> ### Author Rebuttal · Authors · 2024-08-03
>
> We thank the reviewer for their positive review and their helpful comments! Please see responses to individual questions below.
>
> > If we already have the strong convexity and assume the L is differentiable, can we compute the gradient, run the gradient descent, and reduce the problem to the convex optimization?
>
> We would like to distinguish between strong convexity of the loss function and strong convexity with respect to a parameter belonging to some subset of Euclidean space.  Note that taking a gradient and running gradient descent does not make sense in the context of our paper as we are working with arbitrary function classes that do not necessarily live in a space where it is meaningful to take gradients.  The loss function is a function on real numbers and thus strong convexity does make sense.  By way of analogy, if we are training a neural network with square loss, the loss function itself may be strongly convex with respect to the *output* of the neural network, but is certainly *not* convex with respect to the parameters.
>
> > What is the main motivation for considering Gaussian complexity rather than the VC dimension? Can the results from previous studies be extended to Gaussian complexity?
>
> Gaussian complexity is a standard notion of complexity in learning theory and is equivalent to Rademacher complexity up to polylogarithmic factors.  There are many known relationships between these notions of complexity and VC dimension, but Gaussian complexity is strictly more general as VC dimension requires binary-valued function classes.  To the best of our knowledge, our work is the first to produce oracle-efficient algorithms in the semi-private setting that are provably effective in this level of generality.
>
> > There is a lack of discussion on sample complexity. For instance, Theorem 2 requires $n \geq \Omega(1/\alpha^{14})$. It is helpful to discuss the origin of this term.
>
> We agree that the precise polynomial in the sample complexity guarantee for our most general algorithm is higher than what is information theoretically optimal in the absence of computational constraints.  Certainly in an information theoretic sense sample complexity can be improved simply by constructing an  $\epsilon$-net on the function class using the public data as we discuss in Lines 39-46, although the resulting algorithm is impractical. On the other hand, our guarantees are the first to hold for oracle-efficient algorithms in this generality. We leave as an interesting open question the challenge of designing oracle-efficient algorithms with improved sample complexity.
>
> >  What is the state-of-the-art (SOTA) performance without public data? Does the proposed approach with public data outperform current methods in terms of sample complexity?
>
> By works such as [1] and [2], we know that private learning in the absence of public data is not always possible.  To be more precise, any class with infinite Littlestone dimension but finite VC dimension (such as that of linear thresholds in Euclidean space) is PAC learnable but not privately learnable. These classes are privately learnable with public data however, and our paper provides oracle-efficient algorithms for learning them.
>
> [1] Noga Alon, Roi Livni, Maryanthe Malliaris, and Shay Moran. Private PAC learning implies finite littlestone dimension. In Moses Charikar and Edith Cohen, editors, Proceedings of the 51st Annual ACM SIGACT Symposium on Theory of Computing, STOC 2019, Phoenix, AZ, USA, June 23-26, 2019, pages 852–860. ACM, 2019.
>
> [2] Mark Bun, Kobbi Nissim, Uri Stemmer, and Salil Vadhan. Differentially private release and learning of threshold functions. In 2015 IEEE 56th Annual Symposium on Foundations of Computer Science, pages 634–649. IEEE, 2015.

---

### Official Review · Reviewer_CX9N · 2024-07-09

**Soundness:** 3
**Presentation:** 4
**Contribution:** 4
**Rating:** 8
**Confidence:** 3

**Summary:**

This paper considers the setting of differentially private learning when there is some amount of public data available. A downside of existing algorithms is that they generally use the public data to build a cover which ends up being inefficient. A natural question is to design more efficient algorithms that do make use of public data.

The main result of this paper is to show that when the Gaussian complexity of the hypothesis class is small then there is a poly-time algorithm for the problem. Furthermore, if the hypothesis class is well-structured (here they use convexity as the structure) then there is an FTRL-type algorithm that is significantly more efficient and also works under pure DP. The authors also mention that their results hold in the setting where the public data and private data may be different and they quantify this via the ratio between the cdf's.

The actual algorithms themselves actually appear quite simply and practical. In particular, one defines either a noisy or regularized ERM problem and then solve another noisy ERM problem.

**Strengths:**

Overall, I think this a solid contribution to the DP literature. It considers the reasonably well-studied problem of private learning with public data but provides new and simple algorithms for the setting. While I have only looked briefly at the proof techniques, they seem interesting and I think researchers working in this area will be interested to learn about them (e.g. the use of FTRL and anti-concentration type of arguments).

**Weaknesses:**

No weaknesses to discuss from my side.

**Questions:**

- In Theorem 1, I am not sure what $d$ refers to. Is it a dimension or it refers to something else? In particular, is there a need for the square root since you just write poly after? Also, why not use $\mathcal{G}$ for the Gaussian complexity here as you do later in the paper?
- I wonder if the authors have a sense what about what happens if their CDF ratio assumption in Line 73 fails to hold everywhere but does hold in all except maybe $\delta$ fraction of the time. I'm thinking in particular if maybe the public data has an exponential tail whereas the private data has a Gaussian tail (or vice-versa). So in particular, the distributions might be fairly similar in most places except in the tail where the ratio can be wildly different. Is this captured in the remark after Definition 3 in line 149?

**Limitations:**

No concerns from me.

---

> ### Author Rebuttal · Authors · 2024-08-03
>
> We thank the reviewer for their positive review and their helpful comments! Please see responses to individual questions below.
>
> > In Theorem 1, I am not sure what $d$ refers to. Is it a dimension or it refers to something else? In particular, is there a need for the square root since you just write poly after? Also, why not use $\mathcal G$ for the Gaussian complexity here as you do later in the paper?
>
> Thank you for pointing out this confusing passage.  Here the $d$ is supposed to be the VC dimension.  We agree that this would be substantially clearer if we replaced it by the Gaussian complexity and will do this in the camera ready version.
>
> >I wonder if the authors have a sense what about what happens if their CDF ratio assumption in Line 73 fails to hold everywhere but does hold in all except maybe $\delta$ fraction of the time. I'm thinking in particular if maybe the public data has an exponential tail whereas the private data has a Gaussian tail (or vice-versa). So in particular, the distributions might be fairly similar in most places except in the tail where the ratio can be wildly different. Is this captured in the remark after Definition 3 in line 149?
>
> This is a good question.  The specific example raised is not technically covered by the remark in line 149 because the relevant $f$-divergences might not be bounded between exponential and gaussian tails.  Note that the smoothness assumption is only used in ensuring that the perturbed ERMs remain good learners on the marginal distribution over private features and thus any assumption that ensures this will lead to a bound on how well our algorithms learn.  In the case of the $\delta$ fraction not having bounded Radon Nikodym derivative, as long as the loss function is uniformly bounded, minor modification to our analysis ensures that we simply pay a n additive term of $\delta$ in the loss.  We emphasize that the smoothness assumption is not required to ensure privacy of the algorithm and thus our privacy guarantees hold unconditionally.

---

> > ### Comment · Reviewer_CX9N · 2024-08-13
> >
> > Thanks for the response. I will maintain my evaluation.

---

### Official Review · Reviewer_3FxP · 2024-07-10

**Soundness:** 4
**Presentation:** 4
**Contribution:** 3
**Rating:** 7
**Confidence:** 3

**Summary:**

This paper studies the problem of semi-private learning. In this setting, both public and private data are given while the learning algorithm only has to satisfy differential privacy with respect to the private part. Previous work has showed that leveraging the sample complexity of private learning can be improved by leveraging public data, though the methods are computationally inefficient.

In this work, the authors propose the first oracle-efficient algorithm for this task, which requires only a polynomial number of calls of  an ERM oracle. Furthermore, they consider the special cases where the functions are convex or the problem is binary classification and obtain algorithms with improved sample complexity and number of oracle calls, and can satisfy strong privacy property (pure DP).

**Strengths:**

1. The paper is the first to give an oracle-efficient semi-private learning algorithm. Oracle efficiency could be useful in practice due to the fact that many algorithms have large theoretical bounds but work quite well in practice. An oracle-efficient algorithm can directly use them as black-boxes. Previous work, though attains better sample complexity, fails to exploit such a fact.
2. The algorithms work as long as the distribution of private data is smooth w.r.t. that of public data, i.e., the two distributions are not required to be exactly the same.
3. The algorithms are simple and easy to implement.

**Weaknesses:**

1. The resulting sample complexity, though being polynomial, is much higher than previous work.
2. The algorithm requires calls to an ERM oracle. For many tasks, one may only be able to find an approximate minimizer with a small excess risk. It is not clear whether the conclusion still holds if we only have an approximate ERM oracle but not an exact one.

**Questions:**

1. The algorithms construct semi-private learner using ERM oracles. Is it possible to construct it directly using non-private learners (which may not minimize the empirical risk)?
2. On line 64, page 2, it is said that "Prior work of Bassily et al.[2018] gave an oracle-efficient algorithm in this setting with somewhat better sample complexity". Can you elaborate more on this? It looks like their work only guarantees an error of $c\cdot OPT + \alpha$ in the agnostic setting, which is weaker than the result obtained in this paper.

**Limitations:**

Discussed.

---

> ### Author Rebuttal · Authors · 2024-08-03
>
> We thank the reviewer for their positive review and their helpful comments! Please see responses to individual questions below.
>
> >  The resulting sample complexity, though being polynomial, is much higher than previous work.
>
> We agree that the precise polynomial in the sample complexity guarantee for our most general algorithm is higher than what is information theoretically optimal in the absence of computational constraints.  On the other hand, our guarantees are the first to hold for *oracle-efficient* algorithms in this generality. We leave as an interesting open question the challenge of designing oracle-efficient algorithms with improved sample complexity.
>
> > The algorithm requires calls to an ERM oracle. For many tasks, one may only be able to find an approximate minimizer with a small excess risk. It is not clear whether the conclusion still holds if we only have an approximate ERM oracle but not an exact one.
>
> This is an excellent point that we will clarify in the camera ready version.  We note that our analysis in the case of Theorem 2 (corresponding to Algorithm 2) can be modified slightly to handle the case of approximate minimizers.  Indeed, the learning guarantees proceed virtually unchanged.   For the privacy guarantees, observe that privacy follows directly from stability of the first stage estimator $\overline f$ in the respective algorithms.  By inspecting the proof of Theorem 5 we may see that a similar conclusion holds for approximate minimizers.  A quantitatively weaker version of this result that allows for such approximate minimizers can be found in [1].  In the case of the other two algorithms studied, we believe that similar minor modifications to the current analysis will yield similarly graceful weakening of our privacy guarantees as the additive error in the approximate ERM oracle increases.
>
> > The algorithms construct semi-private learner using ERM oracles. Is it possible to construct it directly using non-private learners (which may not minimize the empirical risk)?
>
> This is a good question.  Note that our analysis really shows that any non-private learner undergoing a similar perturbation as we describe will remain a good learner after applying Algorithm 1, so the question is really if such learners can also be made private.  Here the answer depends on whether the non-private learner in question is stable with respect to the $L^2$ on the empirical measure of the public data; if so then our analysis guarantees privacy, but if not then we cannot ensure privacy.
>
>
> > On line 64, page 2, it is said that "Prior work of Bassily et al.[2018] gave an oracle-efficient algorithm in this setting with somewhat better sample complexity". Can you elaborate more on this? It looks like their work only guarantees an error of $c \cdot \textrm{OPT} + \alpha$ in the agnostic setting, which is weaker than the result obtained in this paper.
>
> Thank you for this point; you are correct and we will update our discussion accordingly.
>
>
>
> [1]  Adam Block, Yuval Dagan, Noah Golowich, and Alexander Rakhlin. Smoothed online learning is as easy as statistical learning. In Conference on Learning Theory, pages 1716–1786. PMLR, 2022.

---

> > ### Comment · Reviewer_3FxP · 2024-08-10
> >
> > Thank you for your response, which has addressed my questions. I decide to maintain my positive score.

---

### Official Review · Reviewer_Fe9c · 2024-07-12

**Soundness:** 3
**Presentation:** 3
**Contribution:** 2
**Rating:** 6
**Confidence:** 3

**Summary:**

The paper investigates oracle-efficient semi-private learning. It provides a general framework for transforming an efficient non-private learner into an oracle-efficient semi-private learner for smooth data distributions. For convex and Lipschitz loss functions, as well as binary classification loss, they instantiate the general framework and provide an implementable algorithm with polynomial oracle calls. The output of the proposed algorithm achieves high accuracy with high probability for any hypothesis class with a bounded VC dimension under smooth data distributions. Although the initial algorithm does not achieve optimal sample complexity, the authors introduce a second algorithm inspired by the Follow-the-Regularized-Leader approach from online learning. This improved algorithm enhances the sample complexity bound and requires only two oracle calls.

**Strengths:**

- The paper is clearly written.
- The proposed methods not only have theoretical guarantees, but are of practical relevance due to its efficiency. For example, algorithm 3 requires only 2 calls of the ERM oracle.
- The results hold under slight distribution shift between the public and private data.

**Weaknesses:**

The sample complexity bounds rely on the assumption of a $\sigma$-smooth data distribution. For instance, in Theorem 6, the public sample complexity is given by $m = \tilde{\Theta}(1/\sigma)$ and $n = \tilde{\Omega}(1/\sigma^{12})$. Could you discuss whether it's possible to improve this dependence on $\sigma$? Additionally, could you elaborate on the necessity of assuming a smooth data distribution? Understanding the implications and possible relaxations of this assumption would be beneficial.

**Questions:**

See weaknesses.

**Limitations:**

I don't identify significant limitations.

---

> ### Author Rebuttal · Authors · 2024-08-03
>
> We thank the reviewer for their positive review and their helpful comments! Please see responses to individual questions below.
>
> > Could you discuss whether it's possible to improve this dependence on $\sigma$.
>
> We believe that it is likely that the precise polynomial dependence on problem parameters can be improved.  Certainly in an information theoretic sense this is possible simply by constructing an $\epsilon$-net on the function class using the public data as we discuss in Lines 39-46, although the resulting algorithm is impractical.  We leave as an interesting open question the challenge of designing oracle-efficient algorithms with improved sample complexity.
>
> > Additionally, could you elaborate on the necessity of assuming a smooth data distribution? Understanding the implications and possible relaxations of this assumption would be beneficial.
>
> Regarding the necessity of assuming a smooth data distribution, at the price of quantitatively worse rates, we can replace the notion of smoothness (uniformly bounded Radon NIkodym derivative) with the weaker notion of bounded $f$-divergence.  This point was addressed in [1] in the context of smoothed online learning and similar techniques could likely be applied here.  Please see the discussion in Lines 149-152 for the relevant remark.  Unfortunately, due to the lower bounds in works such as [2,3], one cannot remove such an assumption altogether.
>
>
>
>
> [1] Adam Block and Yury Polyanskiy. The sample complexity of approximate rejection sampling with applications to smoothed online learning. In Gergely Neu and Lorenzo Rosasco, editors, Proceedings of Thirty Sixth Conference on Learning Theory, volume 195 of Proceedings of Machine Learning Research, pages 228–273. PMLR, 12–15 Jul 2023.
>
> [2] Noga Alon, Roi Livni, Maryanthe Malliaris, and Shay Moran. Private PAC learning implies finite littlestone dimension. In Moses Charikar and Edith Cohen, editors, Proceedings of the 51st Annual ACM SIGACT Symposium on Theory of Computing, STOC 2019, Phoenix, AZ, USA, June 23-26, 2019, pages 852–860. ACM, 2019.
>
> [3] Mark Bun, Kobbi Nissim, Uri Stemmer, and Salil Vadhan. Differentially private release and learning of threshold functions. In 2015 IEEE 56th Annual Symposium on Foundations of Computer Science, pages 634–649. IEEE, 2015.

---

### Official Review · Reviewer_21sV · 2024-07-12

**Soundness:** 4
**Presentation:** 3
**Contribution:** 3
**Rating:** 8
**Confidence:** 4

**Summary:**

This paper addresses the problem of identifying a particular class of functions which can be learnt efficiently with unlabelled public and labeled private samples maintaining privacy with respect to the private samples only. The authors propose an algorithmic framework for privately learning such function classes with public unlabelled data. For the general category of functions, if the expectation of the function with respect to the measure is greater than a certain value, the learning is done in two stages. First they learn the unlabelled data by minimizing the loss function with respect to the labeled private data and then add a noisy gaussian process using the function output of the unlabelled data. Then they simply privatize it by using the private output perturbation method. The algorithm can be further simplified for convex functions by simply using L2 regularizer with respect to the function outputs of the unlabelled data instead of using a Gaussian process in case of the general category. They mainly provide the theoretical guarantees by using tools from existing work on Gaussian anti concentration to bound the stability of the learning algorithm on perturbed datasets in terms of the worst case Gaussian complexity of the function class. Then they use the bound on the stability to get a differential privacy guarantee of the output perturbation method. Finally, they show the fact that the algorithm can learn using standard learning theory arguments. These proposed algorithms are oracle efficient in the sense that they only require a polynomial number of calls to the ERM oracle.

**Strengths:**

1. This work contributes to the area of learning theory and differential privacy in two ways. First it identifies certain classes of functions which can be learnt privately with public data in polynomial calls to the ERM oracle. Secondly, it provides the first algorithm with conditional polynomial time learning and privacy guarantee for these classes of functions.
2. The main novelty in this paper is tightening the result by Block et. Al. [1] to get the stability guarantees for the gaussian process which this algorithm follows paving way for making analysis easier for works which use a similar kind of a framework to learn using auxiliary data.
3. This paper is very self-contained in terms of the content and all the proofs have been reiterated, with the necessary changes, to ensure that a beginner in the area of learning theory and differential privacy can develop a good understanding of the paper.
4. The arguments in this paper are mostly direct and easy to follow, while some more complex arguments like the anti concentration expressions are self contained and do not require any additional references.

[1] Adam Block, Yuval Dagan, Noah Golowich, and Alexander Rakhlin. Smoothed online learning is as easy as statistical learning. In Conference on Learning Theory, pages 1716–1786. PMLR, 2022.

**Weaknesses:**

The main weakness in the paper is not discussing a clear intuition of the role of public data in the algorithm and how it helps to learn when private learning is not possible. The questions regarding the paper mentioned below also follow the same theme. It would be very helpful if the authors can add a specific and detailed discussion on the role of public data answering the questions below.

**Questions:**

The paper is theoretically sound and makes sense from the perspective of getting an oracle efficient algorithm for private learning with public data but the intuition behind algorithm 1 is not completely clear. There is part of the algorithm in which the authors add \omega which is the weighted gaussian sum of the function values of the auxiliary data. I have some questions regarding that
1. Why was there a need to create a gaussian process with respect to the auxiliary data in the first place? How would the analysis have gone differently or learnability would have been hampered simply by running ERM on the standard loss function with respect to the private data?
2. Following up with the first question, can the authors emphasize on the point in the analysis where we got an advantage due to using the public data in the mentioned algorithm? Where did we consider the additional information from the public data in the analysis?
3. Can the authors also give an example or maybe give a specific reference to a paper in which they could mention a particular learning problem which would not be privately learnable but can be learnt with public data? Does this algorithm ensure that the particular example problem can be learnt in polynomial time?

**Limitations:**

The authors have mentioned the limitations of their work multiple times within their text and have provided some open problems to further the direction of research in this topic.

---

> ### Author Rebuttal · Authors · 2024-08-02
>
> We thank the reviewer for their positive review and their helpful comments! Please see responses to individual questions below.
>
> > Why was there a need to create a gaussian process with respect to the auxiliary data in the first place? How would the analysis have gone differently or learnability would have been hampered simply by running ERM on the standard loss function with respect to the private data?
>
> Note that adding the gaussian process serves to ensure privacy and does not help (and could potentially hurt) the learning.  Indeed, our learning analysis is concerned with ensuring that our algorithm does not output a hypothesis that is significantly worse than simply running ERM.  The reason to add the perturbation is to ensure *privacy*.  As discussed in Lines 204-207, our privacy analysis follows by ensuring the first step is stable in $L^2$ of the empirical measure *on the public data*  (Lemma 1) and then boosting that stability guarantee into a privacy guarantee (Lemma 3); if we did not have public data and were to only use ERM, then such an approach is meaningless.  We could still ensure label privacy in this way by dong the same procedure and treating the features as public, unlabelled data, but this would still leak information about these features, thereby precluding a privacy guarantee.
>
> > Following up with the first question, can the authors emphasize on the point in the analysis where we got an advantage due to using the public data in the mentioned algorithm? Where did we consider the additional information from the public data in the analysis?
>
> See previous answer.
>
> > Can the authors also give an example or maybe give a specific reference to a paper in which they could mention a particular learning problem which would not be privately learnable but can be learnt with public data? Does this algorithm ensure that the particular example problem can be learnt in polynomial time?
>
> Two papers that discuss this are [1] and [2], cited in our work.  Note that any class with infinite Littlestone dimension but finite VC dimension (such as that of linear thresholds in Euclidean space) is PAC learnable but not privately learnable.  These classes are privately learnable *with public data* however, and our paper provides oracle-efficient algorithms for learning them.
>
>
>
> [1]  Noga Alon, Roi Livni, Maryanthe Malliaris, and Shay Moran. Private PAC learning implies finite littlestone dimension. In Moses Charikar and Edith Cohen, editors, Proceedings of the 51st Annual ACM SIGACT Symposium on Theory of Computing, STOC 2019, Phoenix, AZ, USA, June 23-26, 2019, pages 852–860. ACM, 2019.
>
> [2] Mark Bun, Kobbi Nissim, Uri Stemmer, and Salil Vadhan. Differentially private release and learning of threshold functions. In 2015 IEEE 56th Annual Symposium on Foundations of Computer Science, pages 634–649. IEEE, 2015.

---

> > ### Comment · Reviewer_21sV · 2024-08-12
> >
> > Thank you for the response to my questions. I will maintain the positive score.

---

### Decision · Program_Chairs · 2024-09-25

**Decision:**

Accept (poster)

**Comment:**

This paper addresses the problem of improving private learning algorithms by leveraging public (unlabelled) data. It proposes the first computationally efficient algorithm(s) that guarantee differential privacy with respect to private samples while also achieving learning guarantees when the private data distribution is close to the public data distribution. The main results include a general framework for efficiently leveraging public data, while specialized algorithms are proposed for convex functions and binary classification tasks (with improved sample complexity bounds).

The reviewers found the paper novel and the contributions solid in both privacy and learning guarantees. The reviewers also found the presentation clear.  As feedback for improvement, the paper could benefit from discussing further the potential improvements and/or trade-offs regarding sample complexity. Also, the paper could benefit from further discussions on how the results generalize beyond the the assumptions of the work (e.g., closeness of public and private data distributions). Overall, the reviewers found the paper novel with clear contributions, and they provided feedback for further improving the quality of the work.